# PAC-Bayes under potentially heavy tails

**Matthew J. Holland**
Institute of Scientific and Industrial Research
Osaka University
matthew-h@ar.sanken.osaka-u.ac.jp

## Abstract

We derive PAC-Bayesian learning guarantees for heavy-tailed losses, and obtain a novel optimal Gibbs posterior which enjoys finite-sample excess risk bounds at logarithmic confidence. Our core technique itself makes use of PAC-Bayesian inequalities in order to derive a robust risk estimator, which by design is easy to compute. In particular, only assuming that the first three moments of the loss distribution are bounded, the learning algorithm derived from this estimator achieves nearly sub-Gaussian statistical error, up to the quality of the prior.

## 1 Introduction

More than two decades ago, the origins of PAC-Bayesian learning theory were developed with the goal of strengthening traditional PAC learning guarantees[1] by explicitly accounting for prior knowledge [17, 12, 5]. Subsequent work developed finite-sample risk bounds for "Bayesian" learning algorithms which specify a distribution over the model [13]. These bounds are controlled using the empirical risk and the relative entropy between "prior" and "posterior" distributions, and hold uniformly over the choice of the latter, meaning that the guarantees hold for data-dependent posteriors, hence the naming. Furthermore, choosing the posterior to minimize PAC-Bayesian risk bounds leads to practical learning algorithms which have seen numerous successful applications [2].

Following this framework, a tremendous amount of work has been done to refine, extend, and apply the PAC-Bayesian framework to new learning problems. Tight risk bounds for bounded losses are due to Seeger [15] and Maurer [11], with the former work applying them to Gaussian processes. Bounds constructed using the loss variance in a Bernstein-type inequality were given by Seldin et al. [16], with a data-dependent extension derived by Tolstikhin and Seldin [18]. As stated by McAllester [14], virtually all the bounds derived in the original PAC-Bayesian theory "only apply to bounded loss functions." This technical barrier was solved by Alquier et al. [2], who introduce an additional error term depending on the concentration of the empirical risk about the true risk. This technique was subsequently applied to the log-likelihood loss in the context of Bayesian linear regression by Germain et al. [9], and further systematized by Bégin et al. [3]. While this approach lets us deal with unbounded losses, naturally the statistical error guarantees are only as good as the confidence intervals available for the empirical mean deviations. In particular, strong assumptions on all of the moments of the loss are essentially unavoidable using the traditional tools espoused by Bégin et al. [3], which means the "heavy-tailed" regime cannot be handled, where all we assume is that a few higher-order moments are finite (say finite variance and/or finite kurtosis). A new technique for deriving PAC-Bayesian bounds even under heavy-tailed losses is introduced by Alquier and Guedj [1]; their lucid procedure provides error rates even under heavy tails, but as the authors recognize, the guarantees are sub-optimal at high confidence levels due to direct dependence on the empirical risk, leading in turn to sub-optimal algorithms derived from these bounds.[2]

In this work, while keeping many core ideas of Bégin et al. [3] intact, using a novel approach we obtain exponential tail bounds on the excess risk using PAC-Bayesian bounds that hold even under heavy-tailed losses. Our key technique is to replace the empirical risk with a new mean estimator inspired by the dimension-free estimators of Catoni and Giulini [7], designed to be computationally convenient. We review some key theory in section 2 before introducing the new estimator in section 3. In section 4 we apply this estimator to the PAC-Bayes setting, deriving a new robust optimal Gibbs posterior. Empirical inquiries into the properties of the new mean estimator are given in section 5. All proofs are relegated to supplementary materials.

## 2 PAC-Bayesian theory based on the empirical mean

Let us begin by briefly reviewing the best available PAC-Bayesian learning guarantees under general losses. Denote by $z_1, \ldots, z_n \in \mathcal{Z}$ a sequence of independent observations distributed according to common distribution $\mu$. Denote by $\mathcal{H}$ a model/hypothesis class, from which the learner selects a candidate based on the $n$-sized sample. The quality of this choice can be measured in a pointwise fashion using a loss function $l : \mathcal{H} \times \mathcal{Z} \to \mathbb{R}$, assumed to be $l \geq 0$. The learning task is to achieve a small risk, defined by $R(h) := \mathbf{E}_\mu\, l(h; z)$. Since the underlying distribution is inherently unknown, the canonical proxy is

$$\widehat{R}(h) := \frac{1}{n} \sum_{i=1}^n l(h; z_i), \quad h \in \mathcal{H}.$$

Let $\nu$ and $\rho$ respectively denote "prior" and "posterior" distributions on the model $\mathcal{H}$. The so-called Gibbs risk induced by $\rho$, as well as its empirical counterpart are given by

$$G_\rho := \mathbf{E}_\rho\, R = \int_\mathcal{H} R(h)\, d\rho(h), \quad \widehat{G}_\rho := \mathbf{E}_\rho\, \widehat{R} = \frac{1}{n} \sum_{i=1}^n \int_\mathcal{H} l(h; z_i)\, d\rho(h).$$

When our losses are almost surely bounded, lucid guarantees are available.

**Theorem 1** (PAC-Bayes under bounded losses [13, 3]). *Assume $0 \leq l \leq 1$, and fix any arbitrary prior $\nu$ on $\mathcal{H}$. For any confidence level $\delta \in (0, 1)$, we have with probability no less than $1 - \delta$ over the draw of the sample that*

$$G_\rho \leq \widehat{G}_\rho + \sqrt{\frac{\boldsymbol{K}(\rho; \nu) + \log(2\sqrt{n}\delta^{-1})}{2n}}$$

*uniformly in the choice of $\rho$.*

Since the "good event" where the inequality in Theorem 1 holds is valid for any choice of $\rho$, the result holds even when $\rho$ depends on the sample, which justifies calling it a posterior distribution. Optimizing this upper bound with respect to $\rho$ leads to the so-called optimal Gibbs posterior, which takes a form which is readily characterized (cf. Remark 13).

The above results fall apart when the loss is unbounded, and meaningful extensions become challenging when exponential moment bounds are not available. As highlighted in section 1 above, over the years, the analytical machinery has evolved to provide general-purpose PAC-Bayesian bounds even under heavy-tailed data. The following theorem of Alquier and Guedj [1] extends the strategy of Bégin et al. [3] to obtain bounds under the weakest conditions we know of.

**Theorem 2** (PAC-Bayes under heavy-tailed losses [1]). *Take any $p > 1$ and set $q = p/(p - 1)$. For any confidence level $\delta \in (0, 1)$, we have with probability no less than $1 - \delta$ over the draw of the sample that*

$$G_\rho \leq \widehat{G}_\rho + \left(\frac{\mathbf{E}_\nu\, |\widehat{R} - R|^q}{\delta}\right)^{\frac{1}{q}} \left(\int_\mathcal{H} \left(\frac{d\rho}{d\nu}\right)^p d\nu\right)^{\frac{1}{p}}$$

*uniformly in the choice of $\rho$.*

For concreteness, consider the case of $p = 2$, where $q = 2/(2 - 1) = 2$, and assume that the variance of the loss is $\mathrm{var}_\mu\, l(h; z)$ is $\nu$-finite, namely that

$$V_\nu := \int_\mathcal{H} \mathrm{var}_\mu\, l(h; z)\, d\nu(h) < \infty.$$

From Proposition 4 of Alquier and Guedj [1], we have $\mathbf{E}_\nu |\widehat{R} - R|^2 \le V_\nu/n$. It follows that on the high-probability event, we have

$$G_\rho \le \widehat{G}_\rho + \sqrt{\frac{V_\nu}{n\,\delta}\left(\int_{\mathcal{H}}\left(\frac{d\rho}{d\nu}\right)^2 d\nu\right)}$$

While the $\sqrt{n}$ rate and dependence on a divergence between $\nu$ and $\rho$ are similar, note that the dependence on the confidence level $\delta \in (0,1)$ is polynomial; compare this with the logarithmic dependence available in Theorem 1 above when the losses were bounded.

For comparison, our main result of section 4 is a uniform bound on the Gibbs risk: with probability no less than $1 - \delta$, we have

$$G_\rho \le \widehat{G}_{\rho,\psi} + \frac{1}{\sqrt{n}}\left(\boldsymbol{K}(\rho;\nu) + \frac{\log(8\pi M_2\delta^{-2})}{2} + M_2 + \nu_n^*(\mathcal{H}) - 1\right) + O\left(\frac{1}{n}\right)$$

where $\widehat{G}_{\rho,\psi}$ is an estimator of $G_\rho$ defined in section 3, $\nu_n^*(\mathcal{H})$ is a term depending on the quality of prior $\nu$, and the key constants are bounds such that for all $h \in \mathcal{H}$ we have $M_2 \ge \mathbf{E}_\mu\, l(h;\boldsymbol{z})^2$. As long as the first three moments are finite, this guarantee holds, and thus both sub-Gaussian and heavy-tailed losses (e.g., with infinite higher-order moments) are permitted. Given any valid $M_2$, the PAC-Bayesian upper bound above can be minimized in $\rho$ based on the data, and thus an optimal Gibbs posterior can also be computed in practice. In section 4, we characterize this "robust posterior."

## 3   A new estimator using smoothed Bernoulli noise

**Notation**   In this section, we are dealing with the specific problem of robust mean estimation, thus we specialize our notation here slightly. Data observations will be $x_1, \ldots, x_n \in \mathbb{R}$, assumed to be independent copies of $x \sim \mu$. Denote the index set $[k] := \{1, 2, \ldots, k\}$. Write $\mathcal{M}_+^1(\Omega, \mathcal{A})$ for the set of all probability measures defined on the measurable space $(\Omega, \mathcal{A})$. Write $\boldsymbol{K}(P, Q)$ for the relative entropy between measures $P$ and $Q$ (also known as the KL divergence; definition in appendix). We shall typically suppress $\mathcal{A}$ and even $\Omega$ in the notation when it is clear from the context. Let $\psi$ be a bounded, non-decreasing function such that for some $b > 0$ and all $u \in \mathbb{R}$,

$$-\log\left(1 - u + u^2/b\right) \le \psi(u) \le \log\left(1 + u + u^2/b\right). \tag{1}$$

As a concrete and analytically useful example, we shall use the piecewise polynomial function of Catoni and Giulini [7], defined by

$$\psi(u) := \begin{cases} u - u^3/6, & -\sqrt{2} \le u \le \sqrt{2} \\ 2\sqrt{2}/3, & u > \sqrt{2} \\ -2\sqrt{2}/3, & u < -\sqrt{2} \end{cases} \tag{2}$$

which for $b = 2$ satisfies (1). Slightly looser bounds hold with $b = 1$ for an analogous procedure using a Huber-type influence function.

**Estimator definition**   We consider a straightforward procedure, in which the data are subject to a soft truncation after re-scaling, defined by

$$\widehat{x} := \frac{s}{n}\sum_{i=1}^{n} \psi\left(\frac{x_i}{s}\right) \tag{3}$$

where $s > 0$ is a re-scaling parameter. Depending on the setting of $s$, this function can very closely approximate the sample mean, and indeed modifying this scaling parameter controls the bias of this estimator in a direct way, which can be quantified as follows. As the scale grows, note that

$$s\psi\left(\frac{x}{s}\right) = x - \frac{x^3}{6s^2} \to x, \quad \text{as } s \to \infty$$

which implies that taking expectation with respect to the sample and $s \to \infty$, in the limit this estimator is unbiased, with

$$\mathbf{E}\left(\frac{s}{n}\sum_{i=1}^{n}\psi\left(\frac{x_i}{s}\right)\right) = \mathbf{E}_\mu\, x - \frac{\mathbf{E}_\mu\, x^3}{6s^2} \to \mathbf{E}_\mu\, x.$$

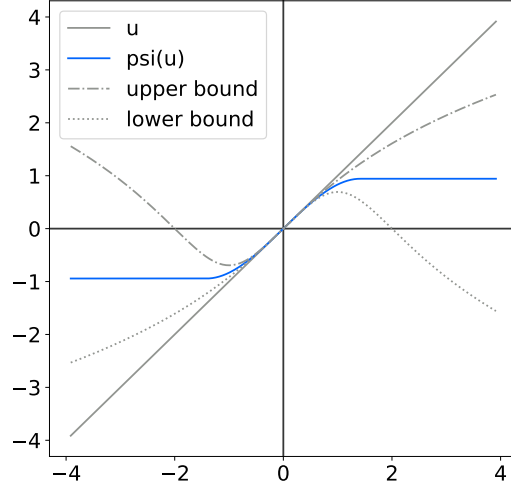

Figure 1: Graph of the Catoni function $\psi(u)$ over $\pm\sqrt{2} \pm 2.5$.

On the other hand, taking $s$ closer to zero implies that more observations will be truncated. Taking $s$ small enough,[3] we have

$$\frac{s}{n}\sum_{i=1}^{n}\psi\left(\frac{x_i}{s}\right) = \frac{2\sqrt{2}s}{3n}\left(|\mathcal{I}_+| - |\mathcal{I}_-|\right),$$

which converges to zero as $s \to 0$. Here the positive/negative indices are $\mathcal{I}_+ := \{i \in [n] : x_i > 0\}$ and $\mathcal{I}_- := \{i \in [n] : x_i < 0\}$. Thus taking $s$ too small means that only the signs of the observations matter, and the absolute value of the estimator tends to become too small.

**High-probability deviation bounds for $\widehat{x}$** We are interested in high-probability bounds on the deviations $|\widehat{x} - \mathbf{E}_\mu\, x|$ under the weakest possible assumptions on the underlying data distribution. To obtain such guarantees in a straightforward manner, we make the simple observation that the estimator $\widehat{x}$ defined in (3) can be related to an estimator with smoothed noise as follows. Let $\epsilon_1, \ldots, \epsilon_n$ be an iid sample of noise $\epsilon \in \{0, 1\}$ with distribution Bernoulli($\theta$) for some $0 < \theta < 1$. Then, taking expectation with respect to the noise sample, one has that

$$\widehat{x} = \frac{1}{\theta}\ \mathbf{E}\left(\frac{s}{n}\sum_{i=1}^{n}\psi\left(\frac{x_i\,\epsilon_i}{s}\right)\right). \tag{4}$$

This simple observation becomes useful to us in the context of the following technical fact.

**Lemma 3.** *Assume we are given some independent data $x_1, \ldots, x_n$, assumed to be copies of the random variable $x \sim \mu$. In addition, let $\epsilon_1, \ldots, \epsilon_n$ similarly be independent observations of "strategic noise," with distribution $\epsilon \sim \rho$ that we can design. Fix an arbitrary prior distribution $\nu$, and consider $f : \mathbb{R}^2 \to \mathbb{R}$, assumed to be bounded and measurable. Write $\mathbf{K}(\rho;\nu)$ for the Kullback-Leibler divergence between distributions $\rho$ and $\nu$. It follows that with probability no less than $1 - \delta$ over the random draw of the sample, we have*

$$\mathbf{E}\left(\frac{1}{n}\sum_{i=1}^{n}f(x_i, \epsilon_i)\right) \leq \int \log \mathbf{E}_\mu \exp(f(x, \epsilon))\,d\rho(\epsilon) + \frac{\mathbf{K}(\rho;\nu) + \log(\delta^{-1})}{n},$$

*uniform in the choice of $\rho$, where expectation on the left-hand side is over the noise sample.*

The special case of interest here is $f(x, \epsilon) = \psi(x\epsilon/s)$. Using (1) and Lemma 3, with prior $\nu = $ Bernoulli($1/2$) and posterior $\rho = $ Bernoulli($\theta$), it follows that on the $1 - \delta$ high-probability event,

uniform in the choice of $0 < \theta < 1$, we have

$$\left(\frac{\theta}{s}\right)\widehat{x} \le \int \left(\frac{\epsilon\,\mathbf{E}_\mu\,x}{s} + \frac{\epsilon^2\,\mathbf{E}_\mu\,x^2}{2s^2}\right)d\rho(\epsilon) + \frac{K(\rho;\nu) + \log(\delta^{-1})}{n} \qquad (5)$$

$$= \frac{\theta\,\mathbf{E}_\mu\,x}{s} + \frac{\theta\,\mathbf{E}_\mu\,x^2}{2s^2} + \frac{1}{n}\left(\theta\log(2\theta) + (1-\theta)\log(2(1-\theta)) + \log(\delta^{-1})\right)$$

where we have used the fact that $\mathbf{E}\,\epsilon^2 = \mathbf{E}\,\epsilon = \theta$ in the Bernoulli case. Dividing both sides by $(\theta/s)$ and optimizing this as a function of $s > 0$ yields a closed-form expression for $s$ depending on the second moment, the confidence $\delta$, and $\theta$. Analogous arguments yield lower bounds on the same quantity. Taking these facts together, we have the following proposition, which says that assuming only finite second moments $\mathbf{E}_\mu\,x^2 < \infty$, the proposed estimator achieves exponential tail bounds scaling with the second non-central moment.

**Proposition 4** (Concentration of deviations). *Scaling with $s^2 = n\,\mathbf{E}_\mu\,x^2/2\log(\delta^{-1})$, the estimator defined in (3) satisfies*

$$|\widehat{x} - \mathbf{E}_\mu\,x| \le \sqrt{\frac{2\,\mathbf{E}_\mu\,x^2\log(\delta^{-1})}{n}} \qquad (6)$$

*with probability at least $1 - 2\delta$.*

*Remark 5.* While the above bound (6) depends on the true second moment, the result is easily extended to hold for any valid upper bound on the moment, which is what will inevitably have to be used in practice.

**Centered estimates** Note that the bound (6) depends on the second moment of the underlying data; this is in contrast to M-estimators which due to a natural "centering" of the data typically have tail bounds depending on the variance [6]. This results in a sensitivity to the absolute value of the location of the distribution, e.g., on a distribution with unit variance and $\mathbf{E}_\mu\,x = 0$ will tend to be much better than a distribution with $\mathbf{E}_\mu\,x = 10^4$. Fortunately, a simple centering strategy works well to alleviate this sensitivity, as follows. Without loss of generality, assume that the first $0 < k < n$ estimates are used for constructing a shifting device, with the remaining $n - k > 0$ points left for running the usual routine on shifted data. More concretely, define

$$\bar{x}_\psi = \frac{\bar{s}}{k}\sum_{i=1}^{k}\psi\left(\frac{x_i}{\bar{s}}\right), \quad\text{where }\bar{s}^2 = \frac{k\,\mathbf{E}_\mu\,x^2}{2\log(\delta^{-1})}. \qquad (7)$$

From (6) in Proposition 4, we have

$$|\bar{x}_\psi - \mathbf{E}_\mu\,x| \le \varepsilon_k := \sqrt{\frac{2\,\mathbf{E}_\mu\,x^2\log(\delta^{-1})}{k}}$$

on an event with probability no less than $1 - 2\delta$, over the draw of the $k$-sized sub-sample. Using this, we shift the remaining data points as $x_i' := x_i - \bar{x}_\psi$. Note that the second moment of this data is bounded as $\mathbf{E}_\mu(x')^2 \le \mathrm{var}_\mu\,x + \varepsilon_k^2$. Passing these shifted points through (3) with analogous second moment bounds used for scaling, we have

$$\widehat{x}' = \frac{s}{(n-k)}\sum_{i=k+1}^{n}\psi\left(\frac{x_i'}{s}\right), \quad\text{where }s^2 = \frac{(n-k)(\mathrm{var}_\mu\,x + \varepsilon_k^2)}{2\log(\delta^{-1})}. \qquad (8)$$

Shifting the resulting output back to the original location by adding and shifting $\widehat{x}'$ back to the original location by adding $\bar{x}_\psi$, conditioned on $\bar{x}_\psi$, we have by (6) again that

$$|(\widehat{x}' + \bar{x}_\psi) - \mathbf{E}_\mu\,x| = |\widehat{x} - \mathbf{E}_\mu(x - \bar{x}_\psi)| \le \sqrt{\frac{2(\mathrm{var}_\mu\,x + \varepsilon_k^2)\log(\delta^{-1})}{n-k}}$$

with probability no less than $1 - 2\delta$ over the draw of the remaining $n - k$ points. Defining the centered estimator as $\widehat{x} = \widehat{x}' + \bar{x}_\psi$, and taking a union bound over the two "good events" on the independent sample subsets, we may thus conclude that

$$\mathbf{P}\left\{|\widehat{x} - \mathbf{E}_\mu\,x| > \varepsilon\right\} \le 4\exp\left(\frac{-(n-k)\varepsilon^2}{2(\mathrm{var}_\mu\,x + \varepsilon_k^2)}\right) \qquad (9)$$

where probability is over the draw of the full $n$-sized sample. While one takes a hit in terms of the sample size, the variance works to combat sensitivity to the distribution location (see section 5 for empirical tests).

# 4 PAC-Bayesian bounds for heavy-tailed data

An import and influential paper due to D. McAllester gave the following theorem as a motivating result. To get started, we give a slightly modified version of his result.

**Theorem 6** (McAllester [12], Preliminary Theorem 2). *Let $\nu$ be a prior probability distribution over $\mathcal{H}$, assumed countable, and to be such that $\nu(h) > 0$ for all $h \in \mathcal{H}$. Consider the pattern recognition task with $z = (x, y) \in \mathcal{X} \times \{-1, 1\}$, and the classification error $l(h; z) = I\{h(x) \neq y\}$. Then with probability no less than $1 - \delta$, for any choice of $h \in \mathcal{H}$, we have*

$$R(h) \leq \frac{1}{n} \sum_{i=1}^{n} l(h; z_i) + \sqrt{\frac{\log(1/\nu(h)) + \log(1/\delta)}{2n}}$$

One quick glance at the proof of this theorem shows that the bounded nature of the observations plays a crucial role in deriving excess risk bounds of the above form, as it is used to obtain concentration inequalities for the empirical risk about the true risk. While analogous concentration inequalities hold under slightly weaker assumptions, when considering the potentially heavy-tailed setting, one simply cannot guarantee that empirical risk is tightly concentrated about the true risk, which prevents direct extensions of such theorems. With this in mind, we take a different approach, that does not require the empirical mean to be well-concentrated.

**Our motivating pre-theorem** The basic idea of our approach is very simple: instead of using the sample mean, bound the off-sample risk using a more robust estimator which is easy to compute directly, and which allows risk bounds even under unbounded, potentially heavy-tailed losses. Define a new approximation of the risk by

$$\widehat{R}_\psi(h) := \frac{s}{n} \sum_{i=1}^{n} \psi\left(\frac{l(h; z_i)}{s}\right), \tag{10}$$

for $s > 0$. Note that this is just a direct application of the robust estimator defined in (3) to the case of a loss which depends on the choice of candidate $h \in \mathcal{H}$. As a motivating result, we basically re-prove McAllester's result (Theorem 6) under much weaker assumptions on the loss, using the statistical properties of the new risk estimator (10), rather than relying on classical Chernoff inequalities.

**Theorem 7** (Pre-theorem). *Let $\nu$ be a prior probability distribution over $\mathcal{H}$, assumed countable. Assume that $\nu(h) > 0$ for all $h \in \mathcal{H}$, and that $m_2(h) := \mathbf{E}\, l(h; z)^2 < \infty$ for all $h \in \mathcal{H}$. Setting the scale in (10) to $s_h^2 = n\, m_2(h)/2\log(\delta^{-1})$, then with probability no less than $1 - 2\delta$, for any choice of $h \in \mathcal{H}$, we have*

$$R(h) \leq \widehat{R}_\psi(h) + \sqrt{\frac{2m_2(h)\left(\log(1/\nu(h)) + \log(1/\delta)\right)}{n}}.$$

*Remark* 8. We note that all quantities on the right-hand side of Theorem 7 are easily computed based on the sample, except for the second moment $m_2$, which in practice must be replaced with an empirical estimate. With an empirical estimate of $m_2$ in place, the upper bound can easily be used to derive a learning algorithm.

**Uncountable model case** Next we extend the previous motivating theorem to a more general result on a potentially uncountable $\mathcal{H}$, using stochastic learning algorithms, as has become standard in the PAC-Bayes literature. We need a few technical conditions, listed below:

1. Bounds on lower-order moments. For all $h \in \mathcal{H}$, we require $\mathbf{E}_\mu\, l(h; z)^2 \leq M_2 < \infty$, $\mathbf{E}_\mu\, l(h; z)^3 \leq M_3 < \infty$.

2. Bounds on the risk. For all $h \in \mathcal{H}$, we require $R(h) \leq \sqrt{nM_2/(4\log(\delta^{-1}))}$.

3. Large enough confidence. We require $\delta \leq \exp(-1/9) \approx 0.89$.

These conditions are quite reasonable, and easily realized under heavy-tailed data, with just lower-order moment assumptions on $\mu$ and say a compact class $\mathcal{H}$. The new terms that appear in our bounds that do no appear in previous works are $\widehat{G}_{\rho,\psi} := \mathbf{E}_\rho\, \widehat{R}_\psi$ and $\nu_n^*(\mathcal{H}) = \mathbf{E}_\nu \exp(\sqrt{n}(R - \widehat{R}_\psi))/\mathbf{E}_\nu \exp(R - \widehat{R}_\psi)$. The former is the expectation of the proposed robust estimator with respect to posterior $\rho$, and the latter is a term that depends directly on the quality of the prior $\nu$.

**Theorem 9.** *Let $\nu$ be a prior distribution on model $\mathcal{H}$. Let the three assumptions listed above hold. Setting the scale in (10) to $s^2 = n\,M_2/2\log(\delta^{-1})$, then with probability no greater than $1 - \delta$ over the random draw of the sample, it holds that*

$$G_\rho \leq \widehat{G}_{\rho,\psi} + \frac{1}{\sqrt{n}}\left(\boldsymbol{K}(\rho;\nu) + \frac{\log(8\pi M_2 \delta^{-2})}{2} + M_2 + \nu_n^*(\mathcal{H}) - 1\right) + O\left(\frac{1}{n}\right)$$

*for any choice of probability distribution $\rho$ on $\mathcal{H}$, since $G_\rho < \infty$ by assumption.*

*Remark* 10. As is evident from the statement of Theorem 9, the convergence rate is clear for all terms but $\nu_n^*(\mathcal{H})/\sqrt{n}$. In our proof, we use a modified version of the elegant and now-standard strategy formulated by Bégin et al. [3]. A glance at the proof shows that under this strategy, there is essentially no way to avoid dependence on $\nu_n^*(\mathcal{H})$. Since the random variable $R - \widehat{R}_\psi$ is bounded over the random draw of the sample and $h \sim \nu$, the bounds still hold and are non-trivial. That said, $\nu_n^*(\mathcal{H})$ may indeed increase as $n \to \infty$, potentially spoiling the $\sqrt{n}$ rate, and even consistency in the worst case. Clearly $\nu_n^*(\mathcal{H})$ presents no troubles if $R - \widehat{R}_\psi \leq 0$ on a high-probability event, but note that this essentially amounts to asking for a prior that on average realizes bounds that are better than we can guarantee for *any* posterior though the above analysis. Such a prior may indeed exist, but if it were known, then that would eliminate the need for doing any learning at all. If the deviations $R - \widehat{R}_\psi$ are truly sub-Gaussian [4], then the $\sqrt{n}$ rate can be easily obtained. However, impossibility results from Devroye et al. [8] suggest that under just a few finite moment assumptions, such an estimator cannot be constructed. As such, here we see a clear limitation of the established PAC-Bayes analytical framework under potentially heavy-tailed data. Since the change of measures step in the proof is fundamental to the basic argument, it appears that concessions will have to be made, either in the form of slower rates, deviations larger than the relative entropy, or weaker dependence on $1/\delta$.

*Remark* 11. Note that while in its tightest form, the above bound requires knowledge of $\mathbf{E}_\mu\, l(h; \boldsymbol{z})^2$, we may set $s > 0$ used to define $\widehat{R}_\psi$ using any valid upper bound $M_2$, under which the above bound still holds as-is, using known quantities. Furthermore, for reference the content of the $O(1/n)$ term in the above bound takes the form

$$\frac{1}{n}\left(2\sqrt{V\log(\delta^{-1})} + \frac{M_3\log(\delta^{-1})}{3M_2\sqrt{n}}\right)$$

where $V$ is an upper bound on the variance $\mathrm{var}_\mu\, l(h; \boldsymbol{z}) \leq V < \infty$ over $h \in \mathcal{H}$.

As a principled approach to deriving stochastic learning algorithms, one naturally considers the choice of posterior $\rho$ in Theorem 9 that minimizes the upper bound. This is typically referred to as the optimal Gibbs posterior [9], and takes a form which is easily characterized, as we prove in the following proposition.

**Proposition 12** (Robust optimal Gibbs posterior)**.** *The upper bound of Theorem 9 is optimized by a data-dependent posterior distribution $\widehat{\rho}$, defined in terms of its density function with respect to the prior $\nu$ as*

$$\left(\frac{d\widehat{\rho}}{d\nu}\right)(h) = \frac{\exp\left(-\sqrt{n}\widehat{R}_\psi(h)\right)}{\mathbf{E}_\nu \exp\left(-\sqrt{n}\widehat{R}_\psi\right)}.$$

*Furthermore, the risk bound under the optimal Gibbs posterior takes the form*

$$G_{\widehat{\rho}} \leq \frac{1}{\sqrt{n}}\left(\log \mathbf{E}_\nu \exp\left(\sqrt{n}\widehat{R}_\psi\right) + \frac{\log(8\pi M_2 \delta^{-1})}{2} + M_2 + \nu_n^*(\mathcal{H}) - 1\right) + O\left(\frac{1}{n}\right)$$

*with probability no less than $1 - \delta$ over the draw of the sample.*

*Remark* 13 (Comparison with traditional Gibbs posterior). In traditional PAC-Bayes analysis [9, Equation 8], the optimal Gibbs posterior, let us write $\widehat{\rho}_{\mathrm{emp}}$, is defined by

$$\left(\frac{d\widehat{\rho}_{\mathrm{emp}}}{d\nu}\right)(h) = \frac{\exp\left(-n\widehat{R}(h)\right)}{\mathbf{E}_\nu \exp\left(-n\widehat{R}\right)}$$

where $\widehat{R}(h) = n^{-1}\sum_{i=1}^n l(h; \boldsymbol{z}_i)$ is the empirical risk. We have $n\widehat{R}$ and $\sqrt{n}\widehat{R}_\psi$, but since scaling in the latter case should be done with $s \propto \sqrt{n}$, so in both cases the $1/n$ factor cancels out. In the special

case of the negative log-likelihood loss, Germain et al. [9] demonstrate that the optimal Gibbs posterior coincides with the classical Bayesian posterior. As noted by Alquier et al. [2], the optimal Gibbs posterior has shown strong empirical performance in practice, and variational approaches have been proposed as efficient alternatives to more traditional MCMC-based implementations. Comparison of both the computational and learning efficiency of our proposed "robust Gibbs posterior" with the traditional Gibbs posterior is a point of significant interest moving forward.

## 5    Empirical analysis

In this section, we use tightly controlled simulations to investigate how the performance of $\widehat{x}$ (cf. (3) and Proposition 4) compares with the sample mean and other robust estimators. We pay particular attention to how performance depends on the underlying distribution family, the value of second moments, and the sample size.

**Experimental setup**    For each experimental setting and each independent trial, we generate a sample $x_1, \ldots, x_n$ of size $n$, compute some estimator $\{x_i\}_{i=1}^n \mapsto \widehat{x}$, and record the deviation $|\widehat{x} - \mathbf{E}_\mu|$. The sample sizes range over $n \in \{10, 20, 30, \ldots, 100\}$, and the number of trials is $10^4$. We draw data from two distribution families, the Normal family with mean $a$ and variance $b^2$, and the log-Normal family, with log-mean $a_{\log}$ and log-variance $b_{\log}^2$, under multiple parameter settings. In particular, we consider the impact of shifting the distribution location over $[-40.0, 40.0]$, with small and large variance settings. Regarding the variance, we have "low," "mid," and "high" settings, which correspond to $b = 0.5, 5.0, 50.0$ in the Normal case, and $b_{\log} = 1.1, 1.35, 1.75$ in the log-Normal case. Over all settings, the log-location parameter of the log-Normal data is fixed at $a_{\log} = 0$. Shifting the Normal data is trivially accomplished by taking the desired $a \in [-40.0, 40.0]$. Shifting the log-Normal data is accomplished by subtracting the true mean (pre-shift) equal to $\exp(a_{\log} + b_{\log}^2/2)$ to center the data, and subsequently adding the desired location.

The methods being compared are as follows: `mean` denotes the empirical mean, `med` the empirical median,[4] `mult_g` is the estimator of Holland [10] using smoothed Gaussian noise, `mult_b` the proposed estimator $\widehat{x}$ defined in (3) using smoothed Bernoulli noise, and finally `mult_bc` the *centered* version of $\widehat{x}$, see the discussion culminating in (9). The latter methods are given access to the true variance or second moment as needed for scaling purposes, and all algorithms are run with confidence parameter $\delta = 0.01$.

**Impact of distribution family**    In Figure 2, we give histograms of the deviations for each method of interest under high variance settings. Colored vertical rules correspond to the error bounds for $\widehat{x}$ under Gaussian noise and Bernoulli noise (bound via Proposition 4), with probability $\delta$. When the standard deviation is not much larger than the mean, we can see substantial improvement over traditional estimators. The bias introduced by the different $\widehat{x}$ choices is clearly far smaller on average than the median, with substantially improved sensitivity to outliers when compared with the mean. The centered version of $\widehat{x}$ has a deviation distribution somewhere between that of the empirical mean and that of the other $\widehat{x}$ choices.

**Impact of distribution location**    In Figure 3 (a), we plot the graph of average/median deviations over trials, taken as a function of the true location $\mathbf{E}_\mu x$. From these results, two clear observations can be made. First, note that the performance of the Gaussian-type (`mult_g`) and Bernoulli-type (`mult_b`) estimators methods tend to differ greatly as a function of the true mean; in particular, we see that the bias of the Gaussian case is far more sensitive to the true location, providing strong evidence for use of our proposed Bernoulli version, which is less expensive, essentially uniformly better than the Gaussian version (as we would expect from the tighter bounds), with error growing slower as a function of the true mean value. Second, the fact that the centering procedure works very well to mitigate the effect of the second moment value is lucid, also a price is paid in overall accuracy due to the naive sample-splitting technique discussed used.

**Impact of sample size**    In Figure 3 (b), we show the graph of average/median deviations taken over all trials, viewed as a function of the sample size $n$. The most distinct observation that can be made

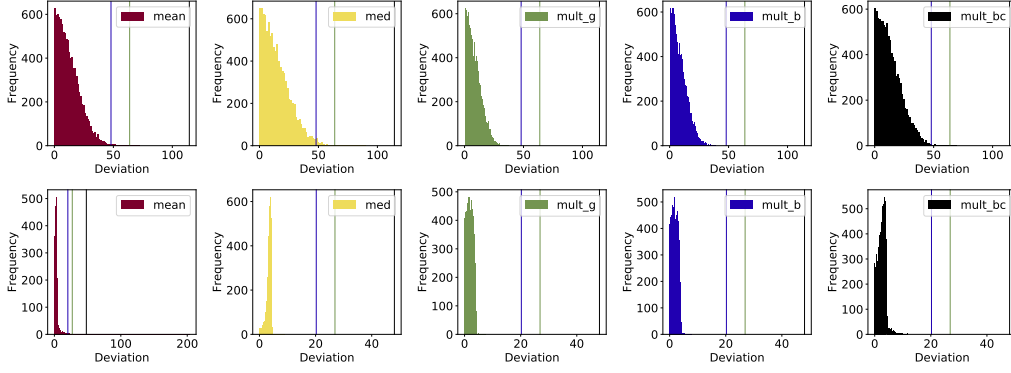

Figure 2: Histograms of deviations $|\widehat{x} - \mathbf{E}_\mu x|$ for different distributions and estimators, with accompanying error bounds. Sample size is $n = 10$. Distributions centered such that mean is equal to "low" level standard deviation. Top: Normal data. Bottom: log-Normal data.

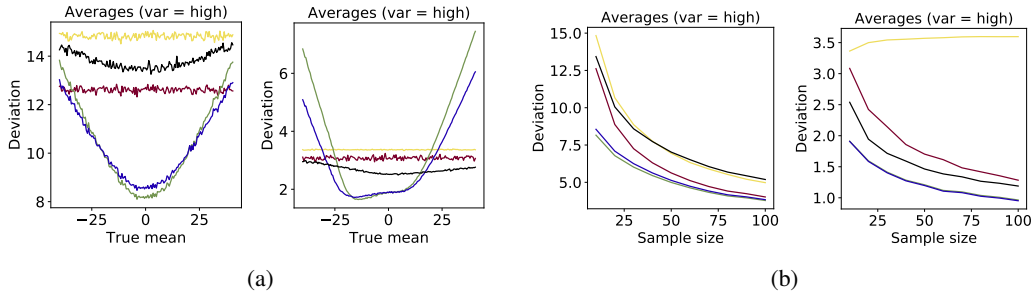

Figure 3: (a) Deviations $|\widehat{x} - \mathbf{E}_\mu x|$ as a function of the true mean $\mathbf{E}_\mu x$. (b) Deviations $|\widehat{x} - \mathbf{E}_\mu x|$ as a function of the sample size $n$. In both sub-figures, left is Normal data, right is log-Normal data.

here is that the estimator $\widehat{x}$ (3) considered here has learning efficiency which is far superior to the empirical mean and median, though as expected the centered version of $\widehat{x}$ has poorer efficiency, a direct result of the sample-splitting scheme used in its definition. As discussed before, this comes with the caveat that the mean cannot be too much larger than the standard deviation; when the second moment is exceedingly large, this leads to a rather large bias as seen in Figure 3 (a) previously.

## 6   Conclusions

The main contribution of this paper was to develop a novel approach to obtaining PAC-Bayesian learning guarantees, which admits deviations with exponential tails under weak moment assumptions on the underlying loss distribution, while still being computationally amenable. In this work, our chief interest was the fundamental problem of obtaining strong guarantees for stochastic learning algorithms which can reflect prior knowledge about the data-generating process, from which we derived a new robust Gibbs posterior. Moving forward, a deeper study of the statistical nature of this new stochastic learning algorithm, as well as computational considerations to be made in practice are of significant interest.

**Acknowledgments**

This work was partially supported by the JSPS KAKENHI Grant Number 18H06477.

## Footnotes

[1] PAC: Probably approximately correct [19].

[2] See work by Catoni [6], Devroye et al. [8] and the references within for background on the fundamental limitations of the empirical mean for real-valued random variables.

[3]More precisely, taking $s \leq \min\{|x_i| : i \in [n]\}/\sqrt{2}$.

[4]After sorting, this is computed as the middle point when $n$ is odd, or the average of the two middle points when $n$ is even.

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
