[Supplementary Material]

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

# A  Technical appendix

## A.1  Preparatory results

**Relative entropy**   Here we recall the basic notions of the relative entropy, or Kullback-Leibler divergence, between two probability distributions. Consider $P$ and $Q$, both defined over a finite space $\Omega$. The relative entropy of $P$ from $Q$ is defined

$$\boldsymbol{K}(P;Q) := \sum_{\omega \in \Omega} P(\omega) \log \left( \frac{P(\omega)}{Q(\omega)} \right), \tag{11}$$

where this definition clearly includes the possibility that $\boldsymbol{K}(P;Q) = \infty$, which occurs only when $Q$ assigns zero probability to an element that $P$ assigns positive probability to.

More generally, when $\Omega$ is potentially uncountably infinite, consider two probabilities $P$ and $Q$ on the measurable space $(\Omega, \mathcal{A})$, where $\mathcal{A}$ is an appropriate $\sigma$-algebra.[5] In this case, the relative entropy is defined

$$\boldsymbol{K}(P;Q) := \int_{\Omega} \log \left( \frac{dP}{dQ} \right) dP, \qquad P \ll Q \tag{12}$$

where $dP/dQ$ denotes the Radon-Nikodym derivative of $P$ with respect to $Q$, typically called the density of $P$ with respect to $Q$. The basic underlying technical assumption, denoted $P \ll Q$, is that $P$ be absolutely continuous with respect to $Q$, meaning that $P(A) = 0$ whenever $Q(A) = 0$, for $A \in \mathcal{A}$. In the event that $P \ll Q$ does not hold, by convention we define $\boldsymbol{K}(P;Q) := \infty$. Recall that the Radon-Nikodym theorem guarantees that when $P \ll Q$, there exists a measurable function $g \geq 0$ such that

$$P(A) = \int_A g \, dQ, \qquad A \in \mathcal{A}.$$

This function $g$ is unique in the sense that if there exists another $f$ satisfying the above equality, then $f = g$ almost everywhere $[Q]$. This uniqueness justifies using the notation $dP/dQ$, and calling this function *the* density of $P$ (rather than *a* density of $P$).

**Lemma 14** (Chain rule). *On measure space $(\Omega, \mathcal{A}, Q)$, let $g \geq 0$ be a Borel-measurable function, and define measure $P$ by*

$$P(A) = \int_A g \, dQ, \qquad A \in \mathcal{A}.$$

*For any Borel-measurable function $f$ on $\Omega$, it follows that*

$$\int_{\Omega} f \, dP = \int_{\Omega} fg \, dQ.$$

*Proof.* See section 2.2, problem 4 of Ash and Doleans-Dade [4].  □

**Lemma 15** (Non-negativity of relative entropy). *For any probabilities $P$ and $Q$, we have $\boldsymbol{K}(P;Q) \geq 0$.*

*Proof of Lemma 15.* If $P \ll Q$ does not hold, then $\boldsymbol{K}(P;Q) = \infty$ and non-negativity follows trivially. As for the case of $P \ll Q$, we begin with the basic logarithmic inequality $x < (1 + x) \log(1 + x)$ for any $x > -1$ [1]. We thus have $x - 1 < x \log(x)$ for any $x > 0$. Using this inequality and the chain rule (Lemma 14), we have

$$\boldsymbol{K}(P;Q) = \mathbf{E}_P \log \frac{dP}{dQ}$$

$$= \mathbf{E}_Q \frac{dP}{dQ} \log \frac{dP}{dQ}$$

$$\geq \mathbf{E}_Q \left( \frac{dP}{dQ} - 1 \right)$$

$$= 0.$$

The final equality uses the Radon-Nikodym theorem.  □

**Lemma 16** (Lower bound on Bernoulli relative entropy)**.** *The relative entropy between Bernoulli$(p)$ and Bernoulli$(q)$ is bounded below by* $\boldsymbol{K}(p;q) \geq 2(p-q)^2$.

*Proof of Lemma 16.* Consider the function $f(p,q)$ defined

$$f(p,q) := \boldsymbol{K}(p;q) - 2(p-q)^2.$$

Fix any arbitrary $p \in (0,1)$, and take the derivative with respect to $q$, noting that

$$\frac{d}{dq} f(p,q) = (-1)(p-q)\left(\frac{1}{q(1-q)} - 4\right).$$

Using the basic fact that $q(1-q) \leq 1/4$ for all $q \in (0,1)$, we have that the factor $(q(1-q))^{-1} - 4$ is non-negative. Thus, the slope is negative when $p > q$, postive when $p < q$, and zero when $p = q$. Thus this is the only minimum of the function in $q$. Note that $f(p,p) = 0$, and so for all $q \in (0,1)$ it follows that $f(p,q) \geq 0$. This holds for any choice of $p$ as well, implying the desired result by the definition of $f$. $\qquad\square$

**Lemma 17** (Chernoff bound for Bernoulli data)**.** *Let* $x_1, \ldots, x_n$ *be independent and identically distributed random variables, taking values* $x \in \{0,1\}$. *Write* $\bar{x} := n^{-1}\sum_{i=1}^n x_i$ *for the sample mean. The tails of the sample mean deviations can be bounded as*

$$\mathbf{P}\{\bar{x} - \mathbf{E}\,x > \varepsilon\} \leq \exp\left(-2n\varepsilon^2\right)$$
$$\mathbf{P}\{\bar{x} - \mathbf{E}\,x < -\varepsilon\} \leq \exp\left(-2n\varepsilon^2\right)$$

*for any* $0 < \varepsilon < 1 - \mathbf{E}\,x$.

*Proof of Lemma 17.* For random variable $x \sim$ Bernoulli$(\theta)$, recall that using Markov's inequality, for any $t > 0$ we have

$$\begin{aligned}
\mathbf{P}\{X > \varepsilon\} &= \mathbf{P}\{\exp(tX) > \exp(t\varepsilon)\} \\
&\leq \exp(-t\varepsilon)\,\mathbf{E}\,e^{tX} \\
&= \exp(-t\varepsilon)\left(1 - \theta + \theta e^t\right).
\end{aligned}$$

Taking the derivative of this upper bound with respect to $t$ and setting it to zero, we obtain the condition

$$t^*(\varepsilon) = \log\left(\frac{\varepsilon}{\theta}\right)\left(\frac{1-\theta}{1-\varepsilon}\right),$$

where we write $t^*(\varepsilon)$ to emphasize the dependence of $t^*$ on $\varepsilon$. We must have $t^*(\varepsilon) > 0$ for the bounds to hold. The value being passed into the $\log$ function must be greater than one. Fortunately, some simple re-arranging of factors shows that

$$\left(\frac{\varepsilon}{\theta}\right)\left(\frac{1-\theta}{1-\varepsilon}\right) > 1 \iff \varepsilon > \theta.$$

So we have $t^*(\varepsilon) > 0$ whenever $\theta < \varepsilon < 1$. Plugging this in, some algebra shows that

$$\begin{aligned}
\exp(-t^*\varepsilon)\left(1 - \theta + \theta e^{t^*}\right) &= \exp\left((1-\varepsilon)\log\left(\frac{1-\theta}{1-\varepsilon}\right) + \varepsilon\log\left(\frac{\theta}{\varepsilon}\right)\right) \\
&= \exp\left(-\boldsymbol{K}(\varepsilon;\theta)\right)
\end{aligned}$$

where we note that the form given in precisely the relative entropy between Bernoulli$(\varepsilon)$ and Bernoulli$(\theta)$.

Returning to the setting of interest with $x_1, \ldots, x_n$ and the sample mean $\bar{x}$, note that using Markov's inequality again and the iid assumption on the data, we have

$$\begin{aligned}
\mathbf{P}\{\bar{x} > \theta + \varepsilon\} &= \mathbf{P}\left\{\sum_{i=1}^n x_i > n(\theta + \varepsilon)\right\} \\
&\leq \left(\exp(-t(\theta+\varepsilon))\,\mathbf{E}_\mu\, e^{tx}\right)^n.
\end{aligned}$$

Setting $t = t^*(\varepsilon + \theta)$ then, and using a classical lower bound on the relative entropy (Lemma 16), we obtain

$$
\begin{aligned}
\mathbf{P}\{\bar{x} > \theta + \varepsilon\} &\leq \left(\exp\left(-\boldsymbol{K}(\theta + \varepsilon; \theta)\right)\right)^n \\
&\leq \left(\exp\left(-2((\theta + \varepsilon) - \theta)^2\right)\right)^n \\
&= \exp\left(-2n\varepsilon^2\right). 
\end{aligned}
\tag{13}
$$

Note that since $\varepsilon + \theta > \theta$ for all $\varepsilon > 0$, it follows that $t^*(\varepsilon + \theta) > 0$ for all $0 < \varepsilon < 1 - \theta$.

Next we seek a lower bound on $\bar{x} - \theta$, equivalently an upper bound on $-\bar{x} + \theta$. This can be done by essentially the same process. Again for $X \sim \text{Bernoulli}(\theta)$, using Markov's inequality, we have for any $s > 0$ that

$$
\begin{aligned}
\mathbf{P}\{X - \theta < -\varepsilon\} &= \mathbf{P}\{-X > \varepsilon - \theta\} \\
&= \mathbf{P}\{\exp(-sX) > \exp(s(\varepsilon - \theta))\} \\
&\leq \exp(-s(\varepsilon - \theta)) \, \mathbf{E} \, e^{-sX} \\
&= \exp(s(\theta - \varepsilon)) \left(1 - \theta + \theta e^{-s}\right).
\end{aligned}
$$

This is, of course, a rather familiar form. Writing $a = \theta - \varepsilon$, note that the function

$$
\exp(sa)\left(1 - \theta + \theta e^{-s}\right)
$$

is minimized as a function of $s$ at

$$
s^* = \log\left(\frac{1-a}{1-\theta}\right)\left(\frac{\theta}{a}\right),
$$

which analogous to earlier in the proof, satisfies $s^* > 0$ only when $\theta > a = \theta - \varepsilon$, which is to say whenever $\varepsilon > 0$. Keeping with the $a$ notation, note that plugging in $s^*$ to the bound above, we have

$$
\begin{aligned}
\exp(s^*a)\left(1 - \theta + \theta e^{-s^*}\right) &= \exp\left((1-a)\log\left(\frac{1-\theta}{1-a}\right) + a\log\left(\frac{\theta}{a}\right)\right) \\
&= \exp\left(-\boldsymbol{K}(a; \theta)\right),
\end{aligned}
$$

the exact same bound as before. It follows that

$$
\begin{aligned}
\mathbf{P}\{\bar{x} - \theta < -\varepsilon\} &= \mathbf{P}\{-\bar{x} > \varepsilon - \theta\} \\
&= \mathbf{P}\left\{-\sum_{i=1}^{n} x_i > n(\varepsilon - \theta)\right\} \\
&\leq \left(\exp(s(\theta - \varepsilon)) \, \mathbf{E}_\mu \, e^{-sx}\right)^n.
\end{aligned}
$$

Setting $s = t^*$ with $a = \theta - \varepsilon$, in a form analogous to the upper bounds done earlier, we have

$$
\begin{aligned}
\mathbf{P}\{\bar{x} - \theta < -\varepsilon\} &\leq \left(\exp\left(-\boldsymbol{K}(\theta - \varepsilon; \theta)\right)\right)^n \\
&\leq \left(\exp\left(-2((\theta - \varepsilon) - \theta)^2\right)\right)^n \\
&= \exp\left(-2n\varepsilon^2\right).
\end{aligned}
\tag{14}
$$

Taking a union bound over the two "bad events" in (13) and (14), we have

$$
\begin{aligned}
\mathbf{P}\{|\bar{x} - \theta| < -\varepsilon\} &\leq \mathbf{P}\{\bar{x} - \theta < -\varepsilon\} \cup \mathbf{P}\{\bar{x} - \theta > \varepsilon\} \\
&\leq \mathbf{P}\{\bar{x} - \theta < -\varepsilon\} + \mathbf{P}\{\bar{x} - \theta > \varepsilon\} \\
&\leq 2\exp\left(-2n\varepsilon^2\right),
\end{aligned}
$$

concluding the proof. $\qquad\square$

**Fundamental PAC-Bayes identity**  The following identity is fundamental to theoretical PAC-Bayesian analysis, and is a well-known result. Catoni [7, p. 159–160] for example gives a concise proof, but for completeness, we provide a step-by-step proof of this result here. The key elements of the following theorem are the prior $\nu \in \mathcal{M}_+^1$, and candidate posterior $\rho \in \mathcal{M}_+^1$.

**Theorem 18.** *For any measurable function h,*

$$\log \mathbf{E}_\nu \exp(h) = \sup_{\rho \in \mathcal{M}_+^1} \left( \sup_{b \in \mathbb{R}} \mathbf{E}_\rho(b \wedge h) - K(\rho; \nu) \right).$$

*In the special case where h is bounded above, then the above equality simplifies to*

$$\log \mathbf{E}_\nu \exp(h) = \sup_{\rho \in \mathcal{M}_+^1} \left( \mathbf{E}_\rho h - K(\rho; \nu) \right).$$

*Proof of Theorem 18.* The key to this proof is a simple expansion of the relative entropy between an arbitrary $\rho \in \mathcal{M}_+^1$ and a specially modified prior $\nu^*$. This $\nu^*$ is defined in terms of the following requirement on the density function $d\nu^*/d\nu$: almost everywhere $[\nu]$, we must have

$$\left( \frac{d\nu^*}{d\nu} \right)(\omega) = g^*(\omega) := \frac{\exp(h(\omega))}{\mathbf{E}_\nu \exp(h)}.$$

Satisfying this is easy by construction. Just define $\nu^*$ using $g^*$, as

$$\nu^*(A) := \int_A g^* \, d\nu, \qquad A \in \mathcal{A}.$$

Since $g^* \geq 0$, it follows that $\nu^*$ is non-negative, and thus a measure on $(\Omega, \mathcal{A})$. As long as $\exp(h)$ is $\nu$-integrable, we have

$$\int_A g^* \, d\nu = (\mathbf{E}_\nu \exp(h))^{-1} \int_A \exp(h(\omega)) \, d\nu(\omega) \leq 1,$$

and also that $\nu^*(\Omega) = 1$, so $\nu^* \in \mathcal{M}_+^1$. Furthermore, note that $\nu^* \ll \nu$ and $\nu \ll \nu^*$.

Now, before proving all the necessary facts, let us run through the primary step of the argument using the following series of identities, which should be rather intuitive even at first glance:

$$K(\rho; \nu^*) = \mathbf{E}_\rho \log \left( \frac{d\rho}{d\nu^*} \right)$$

$$= \mathbf{E}_\rho \log \left( \frac{d\rho}{d\nu} \frac{d\nu}{d\nu^*} \right) \tag{15}$$

$$= \mathbf{E}_\rho \left( \log \frac{d\rho}{d\nu} + \log \frac{d\nu}{d\nu^*} \right) \tag{16}$$

$$= \mathbf{E}_\rho \left( \log \frac{d\rho}{d\nu} + \log \mathbf{E}_\nu \exp(h) - h \right).$$

When the left-hand side is finite, so is the right-hand side, and they are equal. Furthermore, when the left-hand side is infinite, so is the right-hand side.

To prove the above chain of equalities, first start by writing $g(\omega) = (d\nu^*/d\nu)(\omega)$, and observe that by the chain rule (Lemma 14), we have

$$\int_A \left( \frac{1}{g(\omega)} \right) d\nu^* = \int_A \left( \frac{1}{g(\omega)} \right) g(\omega) \, d\nu(\omega) = \nu(A),$$

for any $A \in \mathcal{A}$. By $\nu \ll \nu^*$ and the Radon-Nikodym theorem, it follows that almost everywhere $[\nu]$, we have

$$\left( \frac{d\nu}{d\nu^*} \right)(\omega) = \frac{1}{g(\omega)} = \frac{\mathbf{E}_\nu \exp(h)}{\exp(h(\omega))}, \tag{17}$$

which justifies writing $d\nu/d\nu^* = 1/(d\nu^*/d\nu)$. Another basic fact using the chain rule (Lemma 14) is that for each $A \in \mathcal{A}$,

$$\int_A \left( \frac{d\rho}{d\nu} \right)(\omega) \left( \frac{\mathbf{E}_\nu \exp(h)}{\exp(h(\omega))} \right) d\nu^*(\omega) = \int_A \left( \frac{d\rho}{d\nu} \right)(\omega) \left( \frac{\mathbf{E}_\nu \exp(h)}{\exp(h(\omega))} \right) \left( \frac{\exp(h(\omega))}{\mathbf{E}_\nu \exp(h)} \right) d\nu(\omega)$$

$$= \int_A \frac{d\rho}{d\nu} \, d\nu$$

$$= \rho(A)$$

$$= \int_A \frac{d\rho}{d\nu^*} \, d\nu^*$$

where the final three equalities follow from the Radon-Nikodym theorem and $\rho \ll \nu$ and $\rho \ll \nu^*$. Taking this basic fact and plugging in (17), we have

$$\int_A \frac{d\rho}{d\nu} \frac{d\nu}{d\nu^*} \, d\nu^* = \int_A \frac{d\rho}{d\nu^*} \, d\nu^*, \qquad A \in \mathcal{A}$$

and then by uniqueness of the density function, that almost everywhere $[\nu^*]$,

$$\frac{d\rho}{d\nu} \frac{d\nu}{d\nu^*} = \frac{d\rho}{d\nu^*}.$$

Since any statement a.e. $[\nu^*]$ holds a.e. $[\rho]$ by $\rho \ll \nu^*$, this proves (15).

The first equality holds from the definition of relative entropy, and with (15) now established, the remaining two equalities follow immediately from (17).

The next step is to show that we can meaningfully write

$$\mathbf{E}_\rho \left( \log \frac{d\rho}{d\nu} + \log \mathbf{E}_\nu \exp(h) - h \right) = \boldsymbol{K}(\rho; \nu) + \log \mathbf{E}_\nu \exp(h) - \mathbf{E}_\rho \, h \qquad (18)$$

in the sense that both sides are well-defined, and take on equal values in $\mathbb{R} \cup \{\infty\}$. To prove this, we would like to use the basic additivity property of Lebesgue integrals [4, Theorem 1.6.3]. First observe that the integrand of the left-hand side is well-defined and equal to $\boldsymbol{K}(\rho; \nu^*)$. We need to show that the right-hand side is also well-defined. The first term $\boldsymbol{K}(\rho; \nu) \geq 0 > -\infty$ by Lemma 15, and thus while it cannot be $-\infty$, it takes values in $\mathbb{R} \cup \{\infty\}$. The remaining term depends on $h$. In the case that $h$ is bounded above, we have that $\mathbf{E}_\rho \, h < \infty$, meaning that the right-hand side of (18) is well-defined, which implies via additivity that both sides of (18) take values in $\mathbb{R} \cup \{\infty\}$, and are equal in both the finite and infinite cases.

Note that when $h$ is not bounded above, this leaves the possibility that $\mathbf{E}_\rho \, h = \infty$, which would lead to the ambiguous $\infty - \infty$ on the right-hand side of (18), spoiling the additivity property.

With the assumption of $h$ bounded above, and re-arranging some terms, we can write

$$\boldsymbol{K}(\rho; \nu^*) = \log \mathbf{E}_\nu \exp(h) - (\mathbf{E}_\rho \, h - \boldsymbol{K}(\rho; \nu)).$$

By non-negativity of the relative entropy (Lemma 15), the left-hand side is minimized when $\rho = \nu^*$, in which case it takes the value $\boldsymbol{K}(\nu^*; \nu^*) = 0$. Note that as $\nu^* \in \mathcal{M}_+^1$, the supremum of the term in parentheses on the right-hand side is achieved at $\rho = \nu^*$. This means we can write

$$\log \mathbf{E}_\nu \exp(h) = \sup_{\rho \in \mathcal{M}_+^1} (\mathbf{E}_\rho \, h - \boldsymbol{K}(\rho; \nu)) \qquad (19)$$

for $h$ bounded above.

To complete the proof, we must consider the case where $h$ is unbounded. As preparation, create a measurable function sequence $(h_k)$ defined by $h_k = b_k \wedge h$, where $(b_k)$ satisfies $b_k \uparrow \infty$ and is increasing. Since we have

$$\lim_{k \to \infty} \exp(h_k(\omega)) = \exp(h(\omega))$$

pointwise in $\omega \in \Omega$, and $h_k \leq h_{k+1} \leq \ldots \leq h$ for any $k$, by the monotone convergence theorem, we have

$$\lim_{k \to \infty} \mathbf{E}_\nu \exp(h_k) = \mathbf{E}_\nu \exp(h),$$

and using the continuity of the log function,

$$\lim_{k \to \infty} \log \mathbf{E}_\nu \exp(h_k) = \log \left( \lim_{k \to \infty} \mathbf{E}_\nu \exp(h_k) \right) = \log \mathbf{E}_\nu \exp(h).$$

This means we can write

$$\log \mathbf{E}_\nu \exp(h) = \sup_{b \in \mathbb{R}} \log \mathbf{E}_\nu \exp(b \wedge h)$$

$$= \sup_{b \in \mathbb{R}} \sup_{\rho \in \mathcal{M}_+^1} (\mathbf{E}_\rho (b \wedge h) - \boldsymbol{K}(\rho; \nu)) \qquad (20)$$

$$= \sup_{\rho \in \mathcal{M}_+^1} \sup_{b \in \mathbb{R}} (\mathbf{E}_\rho (b \wedge h) - \boldsymbol{K}(\rho; \nu)) \qquad (21)$$

$$= \sup_{\rho \in \mathcal{M}_+^1} \left( \sup_{b \in \mathbb{R}} \mathbf{E}_\rho (b \wedge h) - \boldsymbol{K}(\rho; \nu) \right).$$

Since for any $b \in \mathbb{R}$, we have that $b \wedge h \leq b < \infty$, we can use (19), the key identity for the case of bounded functions, which immediately implies (20). Finally, regarding the swap of supremum operations, note that the function of interest is

$$f(\rho, b) = \mathbf{E}_\rho(b \wedge h) - \boldsymbol{K}(\rho; \nu), \qquad (\rho, b) \in \mathcal{M}_+^1 \times \mathbb{R}.$$

For an arbitrary sequence $(\rho_k, b_k)$, observe that for all $k$,

$$f(\rho_k, b_k) \leq \sup_\rho f(\rho, b_k) \leq \sup_b \sup_\rho f(\rho, b)$$

$$f(\rho_k, b_k) \leq \sup_b f(\rho_k, b) \leq \sup_\rho \sup_b f(\rho, b).$$

If $f$ is unbounded on $\mathcal{M}_+^1 \times \mathbb{R}$, then the sequence $(\rho_k, b_k)$ can be constructed such that $f(\rho_k, b_k) \to \infty$ as $k \to \infty$, implying that in both cases the supremum is infinite, so equality holds trivially. On the other hand, when $f$ is bounded above, the sequence can be constructed such that $f(\rho_k, b_k) \to B$, and so the above inequalities imply

$$B = \lim_{k \to \infty} f(\rho_k, b_k) \leq \sup_b \sup_\rho f(\rho, b) \leq B$$

$$B = \lim_{k \to \infty} f(\rho_k, b_k) \leq \sup_\rho \sup_b f(\rho, b) \leq B$$

and thus, as desired, the step to (21) holds. This concludes the chain of equalities and the proof. $\square$

## A.2   Proofs of results in the main text

*Proof of Lemma 3.* Start with the following elementary inequality: if $X$ is a random variable such that $\mathbf{E}\, e^X \leq 1$, then for any $\delta \in (0, 1)$, we have that $X$ exceeds $\log(\delta^{-1})$ with probability no greater than $\delta$. To see this, observe that

$$\mathbf{P}\{X \geq \log(\delta^{-1})\} = \mathbf{P}\{\exp(X) \geq 1/\delta\} = \mathbf{E}\, I\{\delta \exp(X) \geq 1\} \leq \mathbf{E}\, \delta e^X \leq \delta. \qquad (22)$$

Next, we set the function $h$ in Theorem 18 to be a sum of functions depending on both the data and the noise, as

$$h(\epsilon) = \sum_{i=1}^n f(x_i, \epsilon) - n \log \mathbf{E}_\mu \exp(f(x, \epsilon)).$$

Since $f$ is bounded on $\mathbb{R}^2$ by hypothesis, we have that $h$ is also bounded. Using Theorem 18, we have

$$B_0 := \sup_{\rho \in \mathcal{M}_+^1} \left( \mathbf{E}_\rho h(\epsilon) - K(\rho; \nu) \right)$$

$$= \log \mathbf{E}_\nu \left( \frac{\exp\left(\sum_{i=1}^n f(x_i, \epsilon)\right)}{(\mathbf{E}_\mu \exp(f(x, \epsilon)))^n} \right).$$

Next, taking expectation with respect to the sample, observe that

$$\mathbf{E} \exp(B_0) = \mathbf{E} \int \left( \frac{\exp\left(\sum_{i=1}^n f(x_i, \epsilon)\right)}{(\mathbf{E}_\mu \exp(f(x, \epsilon)))^n} \right) \nu(\epsilon)$$

$$= \int \left( \frac{\mathbf{E} \exp\left(\sum_{i=1}^n f(x_i, \epsilon)\right)}{(\mathbf{E}_\mu \exp(f(x, \epsilon)))^n} \right) \nu(\epsilon)$$

$$= 1.$$

The above equalities follow from straightforward algebraic manipulations, independence of the data, and taking the integration over the sample inside the integration over the noise, valid using Fubini's theorem. Applying (22) with $X = B_0$, noting that the only randomness is due to the sample, it holds that for probability at least $1 - \delta$, uniform in the choice of $\rho$, we have

$$\mathbf{E}_\rho h(\epsilon) - K(\rho; \nu) \leq \log(\delta^{-1}).$$

Plugging in the above definition of $h$ and dividing by $n$, we have

$$\frac{1}{n} \sum_{i=1}^n \int f(x_i, \epsilon) \, d\rho(\epsilon) \leq \int \log \mathbf{E}_\mu \exp(f(x, \epsilon)) \, d\rho(\epsilon) + \frac{\boldsymbol{K}(\rho; \nu) + \log(\delta^{-1})}{n}.$$

Finally, since the noise observations are iid, we have

$$\frac{1}{n}\sum_{i=1}^{n}\int f(x_i,\epsilon)\,d\rho(\epsilon) = \frac{1}{n}\sum_{i=1}^{n}\int f(x_i,\epsilon_i)\,d\rho(\epsilon_i)$$
$$= \mathbf{E}\left(\frac{1}{n}\sum_{i=1}^{n}f(x_i,\epsilon_i)\right)$$

with expectation over the noise sample. This equality yields the desired result. $\qquad\square$

*Proof of Proposition 4.* First, note that the upper bound derived from (5) holds uniformly in the choice of $\theta$ on a $(1-\delta)$ high-probability event. Setting $\theta = 1/2$ and solving for the optimal $s > 0$ setting is just calculus. It remains to obtain a corresponding lower bound on $\widehat{x} - \mathbf{E}_\mu x$. To do so, consider the analogous setting of Bernoulli $\nu$ and $\rho$, but this time on the domain $\{-1, 0\}$, with $\rho\{-1\} = \theta$ and $\nu\{-1\} = 1/2$. Using (1) and Lemma 3 again, we have

$$\left(\frac{-\theta}{s}\right)\widehat{x} \le \frac{-\theta\,\mathbf{E}_\mu x}{s} + \frac{\theta\,\mathbf{E}_\mu x^2}{2s^2} + \frac{1}{n}\left(\theta\log(2\theta) + (1-\theta)\log(2(1-\theta)) + \log(\delta^{-1})\right)$$

where we note $\mathbf{E}_\rho \epsilon = -\theta$ and $\mathbf{E}_\rho \epsilon^2 = \mathbf{E}_\rho |\epsilon| = \theta$. This yields a high-probability lower bound in the desired form when we set $\theta = 1/2$, since an upper bound on $-\widehat{x} + \mathbf{E}_\mu x$ is equivalent to a lower bound on $\widehat{x} - \mathbf{E}_\mu x$. However, since we have changed the prior in this case, the high-probability event here need not be the same as that for the upper bound, and as such, we must take a union bound over these two events to obtain the desired final result. $\qquad\square$

*Proof of Theorem 6.* For clean notation, denote the empirical risk as

$$\widehat{R}(h) = \frac{1}{n}\sum_{i=1}^{n}l(h;z_i),\qquad h\in\mathcal{H}.$$

Using a classical Chernoff bound specialized to the case of Bernoulli observations (Lemma 17), we have that for any $h\in\mathcal{H}$, it holds that

$$\mathbf{P}\left\{R(h) - \widehat{R}(h) > \varepsilon\right\} \le \exp\left(-2n\varepsilon^2\right).$$

Rearranging terms, it follows immediately that with probability no less than $1 - \nu(h)\,\delta$, we have

$$R(h) - \widehat{R}(h) \le \varepsilon^*(h) := \sqrt{\frac{\log(1/\nu(h)) + \log(1/\delta)}{2n}}.$$

The desired result follows from a union bound:

$$\mathbf{P}\left\{\exists\, h\in\mathcal{H} \text{ s.t. } R(h) - \widehat{R}(h) > \varepsilon^*(h)\right\} \le \mathbf{P}\bigcup_{h\in\mathcal{H}}\left\{R(h) - \widehat{R}(h) > \varepsilon^*(h)\right\}$$
$$\le \sum_{h\in\mathcal{H}}\mathbf{P}\left\{R(h) - \widehat{R}(h) > \varepsilon^*(h)\right\}$$
$$\le \sum_{h\in\mathcal{H}}\nu(h)\delta$$
$$= \delta.$$

The event on the left-hand side of the above inequality is precisely that of the hypothesis, namely the "bad event" on which the sample is such that the risk $R(h)$ exceeds the given bound for *some* candidate $h\in\mathcal{H}$. $\qquad\square$

*Proof of Theorem 7.* We start by making use of the pointwise deviation bound given in Proposition 4, which tells us that with $(1-2\delta)$ high probability

$$R(h) \le \frac{s}{n}\sum_{i=1}^{n}\psi\left(\frac{l(h;z_i)}{s}\right) + \sqrt{\frac{2m_2(h)\log(\delta^{-1})}{n}}$$

for any pre-fixed $h \in \mathcal{H}$. Replacing $\delta$ with $\nu(h)\delta$ gives the key error level

$$\varepsilon^*(h) := \sqrt{\frac{2m_2(h)\left(\log(1/\nu(h)) + \log(1/\delta)\right)}{n}},$$

and using the union bound argument in the proof of Theorem 6, we have

$$\mathbf{P}\left\{\exists\, h \in \mathcal{H} \text{ s.t. } R(h) - \widehat{R}_\psi(h) > \varepsilon^*(h)\right\} \leq 2\delta.$$

<div style="text-align:right">□</div>

*Proof of Theorem 9.* To begin, let us recall a useful "change of measures" inequality,[6] which can be immediately derived from our proof of Theorem 18. In particular, recall from identity (18) that given some prior $\nu$ and constructing $\nu^*$ such that almost everywhere $[\nu]$ one has

$$\left(\frac{d\nu^*}{d\rho}\right)(h) = \frac{\exp(\varphi(h))}{\mathbf{E}_\nu \exp(\varphi)},$$

it follows that

$$\boldsymbol{K}(\rho; \nu^*) = \mathbf{E}_\rho\left(\log\frac{d\rho}{d\nu} + \log\mathbf{E}_\nu\exp(\varphi) - \varphi\right)$$
$$= \boldsymbol{K}(\rho; \nu) + \log\mathbf{E}_\nu\exp(\varphi) - \mathbf{E}_\rho\,\varphi$$

whenever $\mathbf{E}_\rho\,\varphi < \infty$. In the case where $\mathbf{E}_\rho\,\varphi = \infty$, upper bounds are of course meaningless. Re-arranging, observe that since $\boldsymbol{K}(\rho; \nu^*) \geq 0$, it follows that

$$\mathbf{E}_\rho\,\varphi \leq \boldsymbol{K}(\rho; \nu) + \log\mathbf{E}_\nu\exp(\varphi). \tag{23}$$

This inequality given in (23) is deterministic, holds for any choice of $\rho$, and is a standard technical tool in deriving PAC-Bayes bounds.

We shall introduce a minor modification to this now-standard strategy in order to make the subsequent results more lucid. Instead of $\nu^*$ as just characterized above, define $\nu_n^*$ such that almost surely $[\nu]$, we have

$$\left(\frac{d\nu_n^*}{d\rho}\right)(h) = g(h) := \frac{\exp(\varphi(h))}{\mathbf{E}_\nu \exp(\varphi/c_n)},$$

where $1 \leq c_n < \infty$ is a function of the sample size $n$, which increases monotonically as $c_n \uparrow \infty$ when $n \to \infty$ (e.g., setting $c_n = \sqrt{n}$). To explicitly construct such a measure, one can define it by $\nu_n^*(A) := \int_A g\,d\nu$, for all $A \subset \mathcal{A}$, where $(\mathcal{H}, \mathcal{A})$ is our measurable space of interest. In this paper, we always[7] have $\varphi > -\infty$, implying that $\mathbf{E}_\nu \exp(\varphi) > 0$. Also by assumption, since $R$ is bounded over $h \in \mathcal{H}$, we have $\mathbf{E}_\nu \exp(\varphi) < \infty$, which in turn implies

$$0 < \nu_n^*(\mathcal{H}) = \frac{\mathbf{E}_\nu\exp(\varphi)}{\mathbf{E}_\nu\exp(\varphi/c_n)} < \infty,$$

and so $\nu_n^*$ is a finite measure. Note however that both $\nu_n^*(\mathcal{H}) > 1$ and $\nu_n^*(\mathcal{H}) < 1$ are possible, so in general $\nu_n^*$ need not be a probability measure. By construction, we have $\nu_n^* \ll \nu$. Since $\varphi(h) > -\infty$ for all $h \in \mathcal{H}$, we have that $g > 0$ and thus the measurability of $g$ implies the measurability of $1/g$. Using the chain rule (Lemma 14), it follows that for any $A \in \mathcal{A}$,

$$\int_A \left(\frac{1}{g}\right) d\nu_n^* = \int_A \left(\frac{1}{g}\right)(g)\,d\nu = \nu(A).$$

As such, we have $\nu \ll \nu_n^*$, and by the Radon-Nikonym theorem, we may write $1/g = d\nu/d\nu_n^*$ since such a function is unique almost everywhere $[\nu_n^*]$. As long as $\rho \ll \nu$, which in turn implies $\rho \ll \nu_n^*$, so that with use of the chain rule and Radon-Nikodym, we have

$$\int_A \left(\frac{d\rho}{d\nu}\right)\left(\frac{1}{g}\right) d\nu_n^* = \int_A \left(\frac{d\rho}{d\nu}\right)\left(\frac{1}{g}\right) g\,d\nu = \rho(A) = \int_A \left(\frac{d\rho}{d\nu_n^*}\right) d\nu_n^*.$$

Taking the two ends of this string of equalities, by Radon-Nikodym it holds that

$$\frac{d\rho}{d\nu}\frac{d\nu}{d\nu_n^*} = \frac{d\rho}{d\nu_n^*}$$

a.e. $[\nu_n^*]$, and thus a.e. $[\rho]$ as well. Following the argument of Theorem 18, we have that

$$\boldsymbol{K}(\rho;\nu_n^*) = \boldsymbol{K}(\rho;\nu) + \log \mathbf{E}_\nu \exp(\varphi/c_n) - \mathbf{E}_\rho \varphi.$$

The tradeoff for using $\nu_n^*$ which need not be a probability comes in deriving a lower bound on $\boldsymbol{K}(\nu;\nu_n^*)$. In Lemma 15 we showed how the relative entropy between probability measures is non-negative. Non-negativity does not necessarily hold for general measures, but analogous lower bounds can be readily derived for our special case as

$$\boldsymbol{K}(\rho;\nu_n^*) = \mathbf{E}_\rho \log \frac{d\rho}{d\nu_n^*} = \mathbf{E}_{\nu_n^*} \frac{d\rho}{d\nu_n^*} \log \frac{d\rho}{d\nu_n^*} \geq \mathbf{E}_{\nu_n^*} \left(\frac{d\rho}{d\nu_n^*} - 1\right) = 1 - \nu_n^*(\mathcal{H}),$$

where the last inequality uses the fact that $\rho$ is a probability and $\rho(A) = \int_A (d\rho/d\nu_n^*)\, d\nu_n^*$ for all $A \in \mathcal{A}$. Taking this with our decomposition of $\boldsymbol{K}(\rho;\nu_n^*)$, we have

$$\mathbf{E}_\rho \varphi \leq \boldsymbol{K}(\rho;\nu) + \log \mathbf{E}_\nu \exp\left(\varphi/c_n\right) - 1 + \nu_n^*(\mathcal{H}), \tag{24}$$

which amounts to a revised inequality based on change of measures, analogous to (23).

To keep notation clean, write

$$X(h) := R(h) - \frac{s}{n}\sum_{i=1}^{n}\psi\left(\frac{l(h;\boldsymbol{z}_i)}{s}\right) = R(h) - \widehat{R}_\psi(h)$$

$$m_2(h) := \mathbf{E}_\mu\, l(h;\boldsymbol{z})^2$$

$$v(h) := \mathbf{E}_\mu(l(h;\boldsymbol{z}) - R(h))^2$$

Noting that $X(h)$ is random with dependence on the sample, via Markov's inequality we have

$$\mathbf{E}_\nu\, e^X \leq \frac{\mathbf{E}_n\,\mathbf{E}_\nu\, e^X}{\delta}, \tag{25}$$

with probability no less than $1 - \delta$. Here probability and $\mathbf{E}_n$ are with respect to the sample. Since $\widehat{R}_\psi$ is bounded, as long as $\mathbf{E}_\rho R < \infty$, we have $\mathbf{E}_\rho X < \infty$, which lets us use the change of measures inequality in a meaningful way. Now for $c_n > 0$, observe that we have

$$c_n \mathbf{E}_\rho X = \mathbf{E}_\rho c_n X \leq \boldsymbol{K}(\rho;\nu) + \log \mathbf{E}_\nu \exp\left(X\right) - 1 + \nu_n^*(\mathcal{H})$$
$$\leq \boldsymbol{K}(\rho;\nu) + \log(\delta^{-1}) + \log \mathbf{E}_n\,\mathbf{E}_\nu \exp\left(X\right) - 1 + \nu_n^*(\mathcal{H})$$
$$= \boldsymbol{K}(\rho;\nu) + \log(\delta^{-1}) + \log \mathbf{E}_\nu\,\mathbf{E}_n \exp\left(X\right) - 1 + \nu_n^*(\mathcal{H})$$

with probability no less than $1 - \delta$. The first inequality follows from modified change of measures (24), the second inequality follows from (25), and the final interchange of integration operations is valid using Fubini's theorem [4]. Note that the $1 - \delta$ "good event" depends only on $\nu$ (fixed in advance) and not $\rho$. Thus, the above inequality holds on the good event, uniformly in $\rho$.

It remains to bound $\mathbf{E}_n \exp(cX)$, for an arbitrary constant $c > 0$ (here we will have $c = 1$). Start by breaking up the one-sided deviations as

$$X = R - \widehat{R}_\psi = \left(R - \mathbf{E}_n\,\widehat{R}_\psi\right) + \left(\mathbf{E}_n\,\widehat{R}_\psi - \widehat{R}_\psi\right),$$

writing $X_{(1)} := R - \mathbf{E}_n\,\widehat{R}_\psi$ and $X_{(2)} := \mathbf{E}_n\,\widehat{R}_\psi - \widehat{R}_\psi$ for convenience. We will take the terms $X_{(1)}$ and $X_{(2)}$ one at a time. First, note that the function $\psi$ can be written

$$\psi(u) = \left(u - \frac{u^3}{6}\right)\left(I\{u \leq \sqrt{2}\} - I\{u < -\sqrt{2}\}\right) + \frac{2\sqrt{2}}{3}\left(1 - I\{u \leq \sqrt{2}\} - I\{u < -\sqrt{2}\}\right). \tag{26}$$

Again for notational simplicity, write $L = l(h;\boldsymbol{z})$ and $L_i = l(h;\boldsymbol{z}_i)$, $i \in [n]$, where $h \in \mathcal{H}$ is arbitrary. Write $\mathcal{E}_i^+ := \left\{L_i \leq s\sqrt{2}\right\}$ and $\mathcal{E}_i^- := \left\{L_i < -s\sqrt{2}\right\}$. We are assuming non-negative losses, so that

$L \geq 0$. This means that $I(\mathcal{E}_i^-) = 0$ and $\mathbf{P}\,\mathcal{E}_i^- = 0$. We use this, as well as $1 - \mathbf{P}\,\mathcal{E}_i^+ \geq 0$, in addition to (26) in order to bound the expectation of our estimator $\widehat{R}_\psi$ from below, as follows.

$$
\begin{aligned}
\mathbf{E}_n\,\widehat{R}_\psi &= \frac{s}{n}\sum_{i=1}^n \mathbf{E}_\mu\,\psi\left(\frac{L_i}{s}\right) \\
&= \frac{s}{n}\sum_{i=1}^n \left[\mathbf{E}_\mu\left(\frac{L_i}{s} - \frac{L_i^3}{6s^3}\right)\left(I(\mathcal{E}_i^+) - I(\mathcal{E}_i^-)\right) + \frac{2\sqrt{2}}{3}\left(1 - \mathbf{P}\,\mathcal{E}_i^+ - \mathbf{P}\,\mathcal{E}_i^-\right)\right] \\
&\geq \frac{s}{n}\sum_{i=1}^n \mathbf{E}_\mu\left(\frac{L_i}{s} - \frac{L_i^3}{6s^3}\right)I(\mathcal{E}_i^+) \\
&= \mathbf{E}_\mu\,L\,I\{L \leq s\sqrt{2}\} - \frac{1}{6s^2}\,\mathbf{E}_\mu\,L^3 I\{L \leq s\sqrt{2}\} \\
&= R - \mathbf{E}_\mu\,L\,I\{X > s\sqrt{2}\} - \frac{1}{6s^2}\,\mathbf{E}_\mu\,L^3 I\{L \leq s\sqrt{2}\}.
\end{aligned}
$$

By assumption, we have $\mathbf{E}_\mu\,L^3 I\{L \leq s\sqrt{2}\} \leq \mathbf{E}\,L^3 \leq M_3 < \infty$, implying that this lower bound is non-trivial. Next we obtain a one-sided bound on the tails of the loss by

$$
\begin{aligned}
\mathbf{P}\left\{L > s\sqrt{2}\right\} &= \mathbf{P}\left\{L - R > s\sqrt{2} - R\right\} \\
&\leq \mathbf{P}\left\{|L - R| > s\sqrt{2} - R\right\} \\
&\leq \frac{\mathbf{E}_\mu\,|L - R|^2}{\left(s\sqrt{2} - R\right)^2}.
\end{aligned}
$$

Note that the first inequality makes use of $s\sqrt{2} > R$, which is implied by the bounds assumed on $R$, namely that $1/2 \geq R\sqrt{\log(\delta^{-1})/(nM_2)}$.

Returning to the lower bound on $\widehat{R}_\psi$, using Hölder's inequality in conjunction with the tail bound we just obtained, we get an upper bound in the form of

$$
\begin{aligned}
\mathbf{E}_\mu\,L\,I\{L > s\sqrt{2}\} &= \mathbf{E}_\mu\,|L\,I\{L > s\sqrt{2}\}| \\
&\leq \sqrt{\mathbf{E}_\mu\,L^2\,\mathbf{P}\{L > s\sqrt{2}\}} \\
&\leq \sqrt{\frac{\mathbf{E}_\mu\,L^2\,\mathbf{E}_\mu\,|L - R|^2}{\left(s\sqrt{2} - R\right)^2}}.
\end{aligned}
$$

This means we can now say

$$
\mathbf{E}_n\,\widehat{R}_\psi \geq R - \sqrt{\frac{\mathbf{E}_\mu\,L^2\,\mathbf{E}_\mu\,|L - R|^2}{\left(s\sqrt{2} - R\right)^2}} - \frac{1}{6s^2}\,\mathbf{E}_\mu\,L^3 I\{L \leq s\sqrt{2}\},
$$

which re-arranged and written more succinctly gives us

$$
\begin{aligned}
X_{(1)}(h) &\leq \sqrt{\frac{m_2(h)v(h)}{\left(s\sqrt{2} - R\right)^2}} + \frac{\mathbf{E}_\mu\,|l(h;z)|^3}{6s^2} \\
&\leq \sqrt{\frac{M_2 V}{\left(s\sqrt{2} - R\right)^2}} + \frac{M_3}{6s^2} \\
&= \sqrt{\frac{V\log(\delta^{-1})}{n\left(1 - R\sqrt{\log(\delta^{-1})/(nM_2)}\right)^2}} + \frac{M_3\log(\delta^{-1})}{3M_2 n} \\
&\leq B_{(1)} \\
&:= 2\sqrt{\frac{V\log(\delta^{-1})}{n}} + \frac{M_3\log(\delta^{-1})}{3M_2 n}
\end{aligned}
$$

as desired. The final inequality uses the assumed bound on $R$. Note that this is a deterministic bound, in that it is free of both the choice of $h$ (i.e., random draw from $\nu$ or $\rho$) and the sample, which we are integrating over.

Next, we look at the remaining deviations $X_{(2)} = \mathbf{E}_n \widehat{R}_\psi - \widehat{R}_\psi$. Writing $Y_i := (s/n)\psi(L_i/s)$, we have $X_{(2)} = \sum_{i=1}^n (\mathbf{E}\,Y_i - Y_i)$. Since $0 \le \psi(u) \le 2\sqrt{2}/3$ for $u \ge 0$, and $L \ge 0$, we have that $0 \le Y_i \le 2\sqrt{2}s/(3n)$. It follows from Hoeffding's inequality that for all $\epsilon > 0$, we have

$$\mathbf{P}\left\{X_{(2)} > \epsilon\right\} \le \exp\left(\frac{-2\epsilon^2}{n(2\sqrt{2}s/(3n))^2}\right)$$
$$= \exp\left(\frac{-9\epsilon^2 \log(\delta^{-1})}{2M_2}\right). \tag{27}$$

Note that this bound does not depend on the setting of $\delta \in (0,1)$, which is fixed in advance. Also note that while we are dealing with the sum of bounded, independent random variables, the scaling factor $s \propto \sqrt{n}$ makes it such that these deviations converge to some potentially non-zero constant in the $n \to \infty$ limit, which is why $n$ does not appear in the exponential on the right-hand side.

In any case, we can still readily use these sub-Gaussian tail bounds to control the expectation. Using the classic identity relating the expectation to the tails of a distribution,

$$\mathbf{E}_n \exp\left(cX_{(2)}\right) = \int_0^\infty \mathbf{P}\left\{\exp\left(cX_{(2)}\right) > \epsilon\right\} d\epsilon$$
$$= \int_{-\infty}^\infty \mathbf{P}\left\{\exp\left(cX_{(2)}\right) > \exp(\epsilon)\right\} \exp(\epsilon)\, d\epsilon \tag{28}$$

where the second equality follows using integration by substitution. The right-hand side of (28) is readily controlled as follows. Using (27) above, we have

$$\mathbf{P}\left\{\exp\left(cX_{(2)}\right) > \exp(\epsilon)\right\} = \mathbf{P}\left\{X_{(2)} > \epsilon/c\right\} \le \exp\left(\frac{-\epsilon^2}{2\sigma^2}\right)$$

where we have set $\sigma^2 := c^2 M_2/(9\log(\delta^{-1}))$. The key bound of interest can be compactly written as

$$\mathbf{E}_n \exp\left(cX_{(2)}\right) \le 2\int_{-\infty}^\infty \exp\left(-\frac{\epsilon^2}{2\sigma^2} + \epsilon\right) d\epsilon$$
$$= 2\int_{-\infty}^\infty \exp\left(-\frac{1}{2\sigma^2}\left(\epsilon - \sigma^2\right)^2 + \frac{\sigma^2}{2}\right) d\epsilon$$
$$= 2\sqrt{2\pi}\sigma \exp\left(\frac{\sigma^2}{2}\right) \int_{-\infty}^\infty \frac{1}{\sqrt{2\pi}\sigma} \exp\left(-\frac{1}{2\sigma^2}\left(\epsilon - \sigma^2\right)^2\right) d\epsilon$$
$$= 2\sqrt{2\pi}\sigma \exp\left(\frac{\sigma^2}{2}\right).$$

Note that the first equality uses the usual "complete the square" identity, and the rest follows from basic properties of the Gaussian integral. Filling in the definition of $\sigma$, we have

$$\mathbf{E}_n \exp\left(cX_{(2)}\right) \le 2\sqrt{2\pi}\left(c\sqrt{\frac{M_2}{9\log(\delta^{-1})}}\right) \exp\left(\frac{c^2 M_2}{9\log(\delta^{-1})}\right).$$

The right-hand side of this inequality is free of the choice of $h \in \mathcal{H}$, and thus taking expectation with respect to $\nu$ yields the same bound, i.e., the same bound holds for $\mathbf{E}_\nu \mathbf{E}_n \exp(cX_{(2)})$. Taking the log of this upper bound, we thus may conclude that

$$\log \mathbf{E}_\nu \mathbf{E}_n \exp(cX_{(2)}) \le \frac{1}{2}\left[\log(8\pi M_2 c^2) - \log(9\log(\delta^{-1}))\right] + \frac{c^2 M_2}{9\log(\delta^{-1})}$$
$$\le \frac{1}{2}\log(8\pi M_2 c^2) + c^2 M_2$$

on an event of probability no less than $1 - \delta$. The latter inequality uses $\delta \le \exp(-1/9)$. For the result of interest here, we can let $c = 1$.

Finally, going back to the bound on $c_n \mathbf{E}_\rho X$, we can control the key term by

$$
\begin{aligned}
\log \mathbf{E}_\nu \mathbf{E}_n \exp(X) &= \log \mathbf{E}_\nu \mathbf{E}_n \exp \left( X_{(1)} + X_{(2)} \right) \\
&= \log \mathbf{E}_\nu \left[ \exp \left( X_{(1)} \right) \mathbf{E}_n \exp \left( X_{(2)} \right) \right] \\
&\leq B_{(1)} + \log \mathbf{E}_\nu \mathbf{E}_n \exp \left( X_{(2)} \right) \\
&\leq B_{(1)} + \frac{1}{2} \log(8\pi M_2 c^2) + c^2 M_2.
\end{aligned}
$$

Setting $c_n = \sqrt{n}$, we have

$$
\sqrt{n}\, \mathbf{E}_\rho X \leq \mathbf{K}(\rho; \nu) + \log(\delta^{-1}) + \log \mathbf{E}_\nu \mathbf{E}_n \exp(X) - 1 + \nu_n^*(\mathcal{H})
$$

$$
\leq \mathbf{K}(\rho; \nu) + \log(\delta^{-1}) + 2\sqrt{\frac{V \log(\delta^{-1})}{n}} + \frac{M_3 \log(\delta^{-1})}{3 M_2 n} + \frac{\log(8\pi M_2)}{2} + M_2 - 1 + \nu_n^*(\mathcal{H}).
$$

Dividing both sides by $\sqrt{n}$ yields the desired result. $\qquad\square$

*Proof of Proposition 12.* To keep the notation clean, write $X = X(h) = -\sqrt{n}\widehat{R}_\psi(h)$. Similar to the proof of Theorem 18, we have

$$
\begin{aligned}
\mathbf{K}(\rho; \widehat{\rho}) &= \mathbf{E}_\rho \log \left( \frac{d\rho}{d\widehat{\rho}} \right) \\
&= \mathbf{E}_\rho \log \left( \frac{d\rho}{d\nu} \frac{d\nu}{d\widehat{\rho}} \right) \\
&= \mathbf{E}_\rho \left( \log \frac{d\rho}{d\nu} + \log \mathbf{E}_\nu \exp(X) - X \right) \\
&= \mathbf{K}(\rho; \nu) + \log \mathbf{E}_\nu \exp(X) - \mathbf{E}_\rho X
\end{aligned}
$$

whenever $\mathbf{E}_\rho X < \infty$. Using non-negativity of the relative entropy (Lemma 15), the left-hand side of this chain of equalities is minimized in $\rho$ at $\rho = \widehat{\rho}$. Since $\log \mathbf{E}_\nu \exp(X)$ is free of $\rho$, it follows that

$$
\begin{aligned}
\widehat{\rho} &\in \arg\min_\rho \left( \mathbf{K}(\rho; \nu) + \mathbf{E}_\rho(-1)X \right) \\
&= \arg\min_\rho \left( \frac{\mathbf{K}(\rho; \nu)}{\sqrt{n}} + \mathbf{E}_\rho \widehat{R}_\psi(h) \right) \\
&= \arg\min_\rho \left( \frac{\mathbf{K}(\rho; \nu)}{\sqrt{n}} + \mathbf{E}_\rho \widehat{R}_\psi(h) + C \right)
\end{aligned}
$$

where $C$ is any term which is constant in $\rho$, for example all the terms in the upper bound of Theorem 9 besides $\widehat{G}_{\rho,\psi} + \mathbf{K}(\rho; \nu)/\sqrt{n}$. This proves the result regarding the form of the new optimal Gibbs posterior.

Evaluating the risk bound under this posterior is straightforward computation. Observe that

$$
\begin{aligned}
\mathbf{K}(\widehat{\rho}; \nu) &= \mathbf{E}_{\widehat{\rho}} \log \frac{d\widehat{\rho}}{d\nu} \\
&= \mathbf{E}_{\widehat{\rho}} \left( X(h) - \log \mathbf{E}_\nu \exp(X) \right) \\
&= -\sqrt{n}\, \mathbf{E}_{\widehat{\rho}} \widehat{R}_\psi - \log \mathbf{E}_\nu \exp \left( -\sqrt{n}\widehat{R}_\psi \right) \\
&= \log \mathbf{E}_\nu \exp \left( \sqrt{n}\widehat{R}_\psi \right) - \sqrt{n}\, \mathbf{E}_{\widehat{\rho}} \widehat{R}_\psi.
\end{aligned}
$$

Substituting this into the upper bound of Theorem 9, the robust empirical mean estimate terms cancel, and we have

$$
G_{\widehat{\rho}} := \mathbf{E}_{\widehat{\rho}} R \leq \frac{1}{\sqrt{n}} \left( \log \mathbf{E}_\nu \exp \left( \sqrt{n}\widehat{R}_\psi \right) + \frac{\log(8\pi M_2 \delta^{-2})}{2} + M_2 + \nu_n^*(\mathcal{H}) - 1 \right) + O\left( \frac{1}{n} \right).
$$

$\qquad\square$

## Footnotes

[1] PAC: Probably approximately correct [22].

[2] See work by Catoni [9], Devroye et al. [11] and the references within for background on the fundamental limitations of the empirical mean for real-valued random variables.

[3]More precisely, taking $s \leq \min\{|x_i| : i \in [n]\}/\sqrt{2}$.

[4]After sorting, this is computed as the middle point when $n$ is odd, or the average of the two middle points when $n$ is even.

[5] A certain degree of measure theory is assumed in this exposition, at approximately the level of the first few chapters of Ash and Doleans-Dade [4].

[6]There are other very closely related approaches to this proof. See Tolstikhin and Seldin [21], Bégin et al. [5] for some recent examples. Furthermore, we note that the key facts used here are also present in Catoni [8].

[7]We will only be using $\varphi \propto R - \widehat{R}_\psi$, so this statement holds via $R \geq 0$ and $\|R_\psi\|_\infty < \infty$.