[Reviews · NeurIPS 2019]

Reviewer 1



The paper is quite pleasant to read and results are neatly introduced. The paper does not introduce radically new ideas, but rather elegantly combines existing concepts (the robust mean estimator from Catoni and Giulini, PAC-Bayes bounds from McAllester, Bégin et al., Alquier and Guedj) to obtain a new result: a PAC-Bayes bound with a slow regime ($\mathcal{O}(1/\sqrt{n})$) and with logarithmic confidence, holding for unbounded losses. Overall, the paper makes a significant contribution to the PAC-Bayes literature and I recommend acceptance. Major comments (no particular order): * it might be useful to novice readers to briefly explain why the strategy presented in Theorem 1 (McAllester's bound) fails when the loss is not bounded. * Section 3, the $x_i$s need to be independent for Lemma 3 to hold. The key idea to derive results from Section 4 is to pick $x_i:=\ell(h;z_i)$ but this seems to prevent that hypotheses $h$ may be data-dependent (as this would no longer ensure that the $\ell(h;z_i)$ would be independent). I see this as a limitation of the method, and would be interested in reading the authors' opinion. Lines 47-48 "Denote by $\mathcal{H}$ a model, from which the learner selects a candidate based on the $n$-sized sample" are a bit awkward in that sense, as it is unclear whether the hypotheses are allowed to depend on data or not. * Practical value of $s$: as it is somewhat classical in the PAC-Bayes literature, a fine tuning of the parameters to the algorithm is crucial. While it is true that replacing $m_2$, $M_2$ and therefore $s$ with empirical counterparts yields a computable bound, this is a different bound. Perhaps an additional comment on the control of the approximation might be useful (see also the discussions in Catoni's, Audibert's papers about the tuning of the inverse temperature $\lambda$ in the Gibbs posterior). * The shifting strategy used in Section 3 (line 141 and onwards) depends on a data-splitting scheme and the subsample size $k$ acts as an additional parameter. Why not optimize (9) with respect to $k$ and how does this choice of $k$ is reflected upon the main bound? * I am unsure that the numerical experiments add much value to the paper in their current form, as the setup appears quite limited ($n\leq 100$, two data-generating distributions, actual variance used to compute the estimator, etc.) and does not shed much light on how the bound compares to other bounds in the literature. * The paper is lacking a comment on the computational complexity of the overall method. At the end of the story, it boils down to computing the mean of the (robustified) Gibbs posterior, which the authors' claim is reachable with Variational Bayes or MCMC methods. First, I am not sure VB is more tailored to the robustified Gibbs posterior than it is for the "classical" one, as the robust mean estimator makes the whole distribution likely to be further from, say, a Gaussian, than the empirical mean. In addition, VB particularly makes sense with a squared loss, which is not necessarily the case. Second, while it is true that the robust mean estimator "is easy to compute [by design]" (lines 4-5), it is not easier / cheaper than the empirical mean. I feel the paper should be more clear about where the proposed method stands regarding computational complexity. * The abstract reads: "[we] demonstrate that the resulting optimal Gibbs posterior enjoys much stronger guarantees than are available for existing randomized learning algorithms". I feel this might be a bit of an overclaim: the resulting algorithm (robustified Gibbs posterior) does enjoy a PAC-Bayes bound where the dependency on the confidence $\delta$ is optimal -- but does that always make the bound tighter? As it is, no evidence is provided in the paper to support that claim. A (possibly numerical) more direct comparison between the bound between lines 202 and 203 (bound for the robustified Gibbs posterior) and e.g. the bound from Alquier and Guedj between lines 75 and 76 (which is a special case as their main bound holds in a more general setting), specialized to their optimal Gibbs posterior (Alquier and Guedj 2018, Section 4), would be a start. * The paper is also lacking a comparison with Tolstikhin and Seldin's bound, which also implies a variance-term. Tolstikhin and Seldin (2013), "PAC-Bayesian-empirical-Bernstein inequality", NIPS Minor comments: * $\mathbf{K}(\cdot;\cdot)$ (the Kullback-Leibler divergence) is never defined. * Line 61, I suggest to rephrase "Optimizing this upper bound" as "Optimizing this upper bound with respect to $\rho$" to make it even more clear to the reader. * Replace $l$ by $\ell$ (more pleasant on the eye!) * Line 82, typo: the as --> as the * Line 89, $x_1,\dots,x_n\in\mathbb{R}_+$ as negative values will not be considered in the sequel * Line 109, typo: the the --> the * Line 119, repeated "assume" * References: improper capitalization of Rényi === after rebuttal === I thank the authors for taking the time to address my concerns in their rebuttal, which I find satisfying. After reading other reviews, rebuttal and engaging in discussion, my score remains unchanged. I have a remark on the proof of Theorem 9 though. On page 17 of the supplementary material, right after equation (25) the authors invoke Proposition 4 to control the probabillity of deviation of X (lines 459--460 - I'll call this inequality (*)). In Proposition 4, the estimator $hat{x}$ depends on $s$ which depends on $\delta$. So I suspect bounding $P(exp(cX)>exp(\epsilon))$ uniformly in $\epsilon$ might not be licit as it is. From my understanding, the bound should still be valid for any $\epsilon$ large enough so that the right hand-side of inequality (*) is smaller than $\delta$ -- but then integrating with respect to $\epsilon$ might not be licit. I am very sorry for not spotting this in my initial review -- please correct if this is wrong.

Reviewer 2



To my knowledge, this is original research and the method is new. The authors make crystal clear comparison with previous work, and related papers are adequately cited (maybe except at the end of first paragraph, where only a review paper about variational approximations of the Gibbs posterior is cited, missing many papers that derive learning algorithms directly from minimizing PAC-Bayesian bounds, e.g. in Germain et al. (JMLR 2015), Roy et al. (AISTATS 2016)). All claims in the paper are thoroughly discussed demonstrated (in appendix). I did not check the detail of all the proofs in appendices, but as far as I checked I found no mistakes. The paper is very well written and organized. I appreciated the fact that the authors take the time (and space) to motivate the results, even by providing a so-called "pre-theorem" that is less general than their main one, but helps the reader understand the significance of the new bound. Finally, I have no doubt that the result is significant: the new PAC-Bayesian bound provided in this paper tackles a limitation of many PAC-Bayesian bounds from the luterature, contributing in opening the way to applying this theory to a broader family of problems.

Reviewer 3



The paper is in general well written. According to my reading, this study does bring some new insights into learning in the presence of heavy tails

[Author Response · NeurIPS 2019]

We would like to thank all the reviewers for their feedback, it is very much appreciated. Below, we respond to the chief questions and comments raised, particularly from Reviewer 1.

**Data-dependent** $h$    Regarding the point raised on whether $h$ can be chosen in a data-dependent fashion, the answer is that this is possible. As the reviewer points out, the basic deviation bounds on the estimator use independence, but these guarantees can be made "uniform" over $h \in \mathcal{H}$ using an argument analogous to McAllester's classical pre-theorems. This is in fact what we are trying to show with our pre-theorem 7; the bound holds with high probability uniformly over the choice of $h$, meaning it can be selected by a data-dependent procedure. We can definitely adjust the wording around this pre-theorem to reinforce this point. In addition, we shall be more explicit in the text regarding why the classical approach fails with unbounded losses.

**Tightness of bounds**    Regarding the point raised about when the bounds are actually tighter – this is definitely a valid point. Under weak moment assumptions on the loss distribution, while the dependence of deviation bounds on the confidence level is exponentially better than the case of using the empirical mean, this is certainly not the only factor in the bounds. As a simple example, when compared with variance-dependent bounds that depend on $1/\delta$ linearly, if one takes the confidence level low enough ($\delta$ large enough), and the second moment is larger than the variance, then one could force such bounds to be numerically smaller than ours. That said, our basic hope is that the proposed approach should really shine at high confidence levels, in particular limiting the risk variance across independent samples when the data may be heavy-tailed. Tightly controlled simulations and numerical tests of precisely when different methods realize superior guarantees are a point of interest that we are pursuing at present. To get things started, a direct comparison of our bound with the Alquier and Guedj bound under some restricted assumptions could be added easily. As well, we will revise the text to be more precise regarding computational cost, including the potential extra cost for setting the scale parameter.

**Empirical study**    Regarding the point about computation, as reviewer 1 says, there is definitely "more to the story," and one of our major goals moving forward is to develop a general-purpose computational strategy to make this robust PAC-Bayes approach practical for a large class of learning problems. As the reviewer points out, there is a great line of recent literature on robust mean estimation in high dimensions. In the context of PAC-Bayes, we are currently seeking a more refined understanding of the tradeoffs between statistical performance guarantees, computational cost, and computational error (i.e., when the estimator cannot be computed exactly) that come up when introducing more sophisticated sub-routines, such as those borne of the recent line of work starting with Lugosi and Mendelson in 2017.

**Algorithm parameter settings**    Regarding the scaling and shifting strategies, in the present paper we have left this mostly abstract, with the implication being that one could plug in empirical moment estimates for the former, and just split the sample in half for the latter, but certainly this strategy is by no means optimal, and for practical purposes, it must be handled very carefully. As we continue with numerical tests in this direction, we plan to flesh out the theory in this direction further, accompanied with some empirical insights to back up our approach. Some comments regarding possible strategies can be added to the manuscript to elucidate this point.

Finally, we would also like to thank all the reviewers for raising various minor points to be corrected. These will all be reflected as we revise the manuscript.

[Meta-Review · NeurIPS 2019]

The paper obtains an interesting new PAC-Bayes bound holding for heavy-tailed losses with a logarithmic confidence term by elegantly combining previous concepts. There is currently a lot of interest in the community in such robust estimators, so this result is expected to have a significant impact.